**Seasonal variability of the inorganic carbon system in a large coastal plain estuary**

Andrew Joesoef[1], David L. Kirchman[2], Christopher K. Sommerfield[2], and Wei-Jun Cai[1]

[1]School of Marine Science and Policy, University of Delaware, Newark, DE 19716, USA

[2]School of Marine Science and Policy, University of Delaware, Lewes, DE 19958, USA

*Correspondence to*: Wei-Jun Cai (wcai@udel.edu)

**Abstract.** Carbonate geochemistry research in large estuarine systems is limited. More work is needed to understand how changes in land use activity influence watershed export of organic and inorganic carbon, acids, and nutrients to the coastal ocean. To investigate the seasonal variation of the inorganic carbon system in the Delaware Estuary, one of the largest estuaries along the U.S. east coast, dissolved inorganic carbon (DIC), total alkalinity (TA), and pH were measured along the estuary from June 2013 to April 2015. In addition, DIC, TA, and pH were periodically measured from March to October 2015 in the non-tidal freshwater Delaware, Schuylkill, and Christina rivers over a range of discharge conditions. There were strong negative relationships between river TA and discharge, suggesting that changes in $HCO_3^-$ concentrations reflect dilution of weathering products in the drainage basin. The ratio of DIC to TA, an understudied but important property, was high (1.11) during high discharge and low (0.94) during low discharge, reflecting additional DIC input in the form of carbon dioxide ($CO_2$), most likely from terrestrial organic matter decomposition, rather than bicarbonate ($HCO_3^-$) inputs due to drainage basin weathering processes as well as limited $CO_2$ loss to the atmosphere due to rapid water transit during wet season. Our data further show that elevated DIC in the Schuylkill River is substantially different than that in the Delaware River. Thus, tributary contributions must be considered when attributing estuarine DIC sources to the internal carbon cycle versus external processes such as drainage basin mineralogy, weathering intensity, and discharge patterns. Long-term records in the Delaware and Schuylkill rivers indicate shifts toward higher alkalinity in estuarine waters over time, as has been found in other estuaries world-wide. Annual DIC input flux to the estuary and export flux to the coastal ocean are estimated to be $15.7 \pm 8.2 \times 10^9$ mol C $yr^{-1}$ and $16.5 \pm 10.6 \times 10^9$ mol C $yr^{-1}$, respectively, while net DIC production within the estuary including inputs from intertidal marshes is estimated to be $5.1 \times 10^9$ mol C $yr^{-1}$. The small difference between riverine input and export flux suggest that, in the case of the Delaware Estuary and perhaps other large

coastal systems with long freshwater residence times, the majority of the DIC produced in the estuary by biological processes is exchanged with the atmosphere rather than exported to the sea.

## 1 Introduction

The global carbon cycle involves dynamical processes of carbon exchange among the earth's
atmosphere, land, vegetation, coastal zones, and oceans. Over the past century, human perturbations and land-use changes have significantly modified the transport of carbon across the land-to-ocean continuum and have resulted in imbalances to present-day carbon fluxes and storage reservoirs (Aumont et al., 2001; Cotrim da Cunha et al., 2007; Quinton et al., 2010; Bauer et al., 2013; Regnier et al., 2013). Most of carbon fluxes in inland waters involve inputs from soil-derived
carbon, chemical weathering of carbonate and silicate minerals, wetlands, dissolved carbon in sewage waste, and organic carbon produced by phytoplankton in surface waters (Battin et al., 2009; Tranvik et al., 2009; Regnier et al., 2013; Abril et al., 2014). To balance the influx of carbon, a large fraction is returned to the atmosphere by organic carbon decomposition within inland waters, transported to adjacent waters, buried in freshwater sediments, and in some cases released
as methane ($CH_4$) gas (Downing et al., 2008; Bastviken et al., 2011).

Total alkalinity (TA) is defined as TAlk = $[HCO_3^-]$ + $2[CO_3^{2-}]$ plus all other weak bases that can accept $H^+$ when titrated to the carbonic acid endpoint. Comparably, dissolved inorganic carbon (DIC) is expressed as the sum of all inorganic carbon species ($[CO_2]$, $[HCO_3^-]$, $[CO_3^{2-}]$). In terrestrial aquatic systems, there are three sources of dissolved inorganic carbon. The most
20 important sources are the carbonate and silicate weathering processes as described below:

$$CaCO_3 + CO_2 \rightarrow 2HCO_3^- + Ca^{2+} \qquad (1)$$
$$CaSiO_3 + 2CO_2 + 3H_2O \rightarrow 2HCO_3^- + Ca^{2+} + H_4SiO_4 \qquad (2)$$

In both cases, the amounts of DIC and TA production are equal. Here, $CO_2$ may come from soil organic matter respiration but ultimately it is linked to the atmospheric $CO_2$. Respiration of soil and aquatic organic carbon is another source of $CO_2$, but it does not contribute to TA. Since alkalinity of natural waters is mainly composed of $[HCO_3^-]$ and $[CO_3^{2-}]$, DIC to TA ratios can provide broad insight into the sources of carbon, aquatic pH dynamics and regional carbonate

buffering capacity. Large shifts in seasonal precipitation and weathering rates can significantly affect DIC and TA concentrations (Probst et al., 1992; Cai, 2003; Guo et al., 2008).

Typically, the supply of inorganic carbon by rivers to the coastal ocean is governed by river discharge, weathering intensity, and the geology of the drainage basin (White and Blum, 1995; White, 2003; Guo et al., 2008). The weathering of carbonate and silicate minerals consumes atmospheric $CO_2$ and transports $HCO_3^-$ ions and subsequent cation and anion products into oceanic systems. Eventually, $CO_2$ is released back into the atmosphere via oceanic carbonate sedimentation and volcanic activity (Lerman et al., 2004; Regnier et al., 2013). Guo et al., (2008) found that in the Pearl River estuary DIC and TA values were substantially lower during the wet season (~ 1000 and 700 µmol $kg^{-1}$, respectively) than during the dry season (> 2700 and > 2400 µmol $kg^{-1}$, respectively). They suggested that the much lower DIC and TA values in the wet season were a result of increased river discharge diluting overall production of DIC and TA by weathering and decomposition. Similar results were found in the Mississippi and Changjiang where river $HCO_3^-$ concentration and discharge are negatively correlated (Cai et al., 2008).

As carbon is transported horizontally along the land and ocean continuum, various environmental processes impact the total carbon fluxes between reservoirs. Recent synthesis suggests that a variable but relatively small fraction of $CO_2$ emitted in estuaries is sustained by freshwater inputs while most of the $CO_2$ released is from local net heterotrophy, with the majority of organic carbon inputs stemming from adjacent salt marsh and mangrove ecosystems (Cai 2011; Regnier et al., 2013). These systems are supported by inputs from various autochtonous and allochtonous organic carbon sources, $CO_2$ enriched sediment intertidal waters during ebbing, and high concentrations of dissolved inorganic carbon from inter-tidal and sub-tidal benthic communities (Cai et al., 2003; Neubauer and Anderson, 2003; Wang and Cai, 2004; Ferrón et al., 2007; Chen and Borges, 2009). Terrestrial organic carbon that is transported by large and fast-transit river systems generally bypasses decomposition in estuaries and contributes to respiration along coastal ocean margins (Cai, 2011). Consequently, rapid increases in atmospheric $CO_2$ concentrations may have reduced the amount of $CO_2$ released along ocean margin systems, especially in low latitude zones where a majority of the terrestrial organic carbon is delivered (Cai, 2011).

While there have been several inorganic carbon studies on rapidly transiting large river systems, globally carbonate chemistry research in large estuaries remains limited (Ternon et al., 2000; Cai, 2003; Cai et al., 2004; Cooley et al., 2007; Cai et al., 2008; Liu et al., 2014). Further, the majority

of past estuarine $CO_2$ studies have focused primarily on small estuarine systems (typically within 1 – 100 km in length and less than 10 m in depth) with rapid water transit and short freshwater residence times ($10^{-3}$ – $10^{-1}$ yr) (Chen and Borges, 2009; Cai, 2011; Borges and Abril, 2011, Dürr et al., 2011). Thus, there is an absence of carbonate research in large estuaries with long water residence times. In this study, we investigated the carbonate geochemistry of the Delaware, Schuylkill, and Christina rivers, the three main tributaries of the Delaware Estuary, which is one of the largest estuaries along the U.S. east coast. Using monthly sampling in 2013-2015, we examined how input from multiple tributaries, contrasting geographical settings, and physical mixing processes affect total riverine DIC and TA fluxes, internal net ecosystem production, and overall export flux in a large coastal plain estuary. Using historical and contemporary data collected along the Delaware and Schuylkill rivers, we further explored how tributaries influence regional trends in riverine carbonate chemistry.

## 2 Methods

### 2.1 Study area

The Delaware Estuary is a 215 km long coastal plain estuary that extends from the head of the tidal Delaware River at Trenton, New Jersey to the mouth of the Delaware Bay between Cape Henlopen and Cape May (Fig. 1). The Delaware River provides 50-60% of the total freshwater inflow to the estuary. Based on U.S. Geological Survey (USGS) stream gauging data, the annual mean discharge of the Delaware River at Trenton is 340 $m^3$ $s^{-1}$ (1950-2015). Of the many small rivers in New Jersey, Pennsylvania, and Delaware that flow into the estuary, the Schuylkill River is the largest with an annual mean discharge of 80 $m^3$ $s^{-1}$, whereas the Brandywine-Christina rivers are the smallest gauged tributaries with a combined mean of 20 $m^3$ $s^{-1}$. Together, the Delaware, Schuylkill, and Brandywine-Christina rivers contribute ~ 70% of the total freshwater input to the estuary with the balance sourced mostly by smaller ungauged rivers (Sutton et al., 1996). Freshwater input from municipal wastewater treatment plants is important as well with a discharge around 30 $m^3$ $s^{-1}$. As the tidal freshwater river passes through the industrial Philadelphia region, it transitions to an extensive river estuary and bay system surrounded by intertidal salt marshes. Depending on precipitation and discharge, freshwater residence times in the Delaware Estuary generally range from 40 to 90 days but may exceed 200 days during periods of drought. Circulation in the estuary is influenced by tides, wind, and dynamical interactions between freshwater runoff

from the drainage basin and saltwater inflow from the Atlantic Ocean (Wong and Sommerfield, 2009; Sommerfield and Wong, 2011; Aristizáhal and Chant, 2015).

## 2.2 Field measurements

DIC, TA, and pH were measured along the salinity gradient of the Delaware Estuary on eight cruises: 8-10 June 2013, 17-22 November 2013, 23-24 March 2014, 2-3 July 2014, 27 August to 1 September 2014, 30 October to 2 November 2014, 5 December 2014, and 6 April 2015. However, because stations were different for each cruise, we do not label them in Fig. 1. Water column samples were collected with a SeaBird Electronics 911 (SBE 911) plus CTD rosette system. Discrete underway samples were taken from the outlet of an onboard SeaBird thermosalinograph (SBE-45), which measured underway surface water temperature and salinity. In addition to the eight cruises, DIC, TA, and pH were periodically collected from March to October 2015 from the Delaware, Schuylkill, and Christina rivers (Fig. 1 and 2; Table 1). Instantaneous water discharge data for the Delaware and Schuylkill rivers were available from gauging stations in Trenton, NJ and Philadelphia, PA (USGS gauges 01463500 and 01474500, respectively) (Fig. 2). Discharge data for the Christina River, Brandywine Creek, Red Clay Creek, and White Clay Creek were used to compute total freshwater discharge for the Christina River system (USGS gauges 01478000, 01481500, 01480015, and 01479000).

## 2.3 Analytical methods

DIC and TA samples were filtered through a cellulose acetate filter (0.45 μm) into 250 ml borosilicate bottles, fixed with 100 μl of saturated mercury bichloride solution, preserved at 4°C, and analyzed within two weeks of sample collection (Cai and Wang, 1998; Jiang et al., 2008). DIC was determined via acid extraction by quantifying the released $CO_2$ using an infrared gas analyzer (AS-C3, Apollo SciTech). TA was measured by Gran titration (Gran, 1952) using an open cell semi-automatic titration system (AS-ALK2, Apollo SciTech) (Cai et al., 2010; Huang et al., 2012). Accepted analytical precision based on three repeats was $\pm 2$ μmol kg$^{-1}$, and all measurements were calibrated against certified reference material (provided by A.G. Dickson from Scripps Institution of Oceanography) (Huang et al., 2012). For pH measurements, water samples were collected in glass bottles with a narrow mouth and left in a thermal bath (at 25°) for about 30-60 minutes. pH was then determined onboard using an Orion 3-Star Plus pH Benchtop Meter with a Ross pH electrode (Thermo Fisher Scientific Inc. Beverly, MA, USA) and calibrated using three National

Bureau Standard (NBS) pH buffers of 4.01, 7.00, and 10.01. Note that the narrow mouth of the glass bottle is only slightly larger than the outer diameter of the pH electrode thus preventing $CO_2$ degassing during the analysis. While the analytical precision is $\pm$ 0.005 units, the expected accuracy is probably not better than $\pm$ 0.01 pH units.

## 3 Results

### 3.1 Spatial distributions of DIC and TA

DIC and TA varied greatly in the estuary and with season (975-2015 and 915-2225 µmol kg$^{-1}$, respectively) (Fig. 3). DIC and TA were lowest near zero salinity in the spring and summer when river discharge was strong and were highest in the fall and winter when discharge was weak (Fig. 3). At the bay mouth (S > 30), DIC and TA concentrations remained fairly constant throughout all seasons (1920-1990 and 2095-2180 µmol kg$^{-1}$, respectively). During spring and summer, DIC was reduced while pH (8.0-8.5) was highest in waters with values 15-25 salinity, suggesting biological consumption of $CO_2$ in the mid- to lower bay (Fig. 3). In the fall and winter, DIC and TA generally varied linearly in relation to salinity, although the change in pH was small across the salinity gradient (Fig. 3). At salinity < 2.5, pH decreased greatly, reaching as low as 7.1 in June.

Depending on river discharge conditions, DIC concentrations typically range from about 300 to 1200 µmol kg$^{-1}$ at the head of the estuary at Trenton (Sharp et al., 2009). During the spring and summer surveys when discharge was high, DIC and TA concentrations were about 300 µmol kg$^{-1}$ lower than concentrations in the fall when river discharge was low (Fig. 3). Following a 5-day high discharge event, TA on July 2, 2015 was an average of 410.4 µmol kg$^{-1}$ (Table 1), about 600 µmol kg$^{-1}$ lower than the high TA in the river at low discharge from March to October (Table 1). DIC followed similar patterns. TA also changed greatly at the Schuylkill River. Additionally, when average river discharge in the Schuylkill River was less than 50 m$^3$ s$^{-1}$, TA values exceeded 1500 µmol kg$^{-1}$ (Table 1).

While TA in the Schuylkill River was nearly double that of the Delaware River at Trenton, Delaware River discharge was nearly four-fold greater than the Schuylkill River discharge (Table 1). Moreover, the average discharge in the Delaware River was more than 10-fold greater than discharge of the Christina River (Table 1). Despite mixing from multiple sources, based on the relative discharges of the tributaries, the average riverine TA is predominantly governed by carbonate concentrations in the Delaware River. However, during periods of low discharge, TA

increased significantly at the Schuylkill River (Fig. 4 and Table 1). On September 29, 2015, TA values were as high as the oceanic values at the bay mouth exceeding 2100 µmol kg$^{-1}$. The mixing of high TA from the Schuylkill River may increase TA values at the confluence of the Delaware and Schuylkill River. Slight increases in TA values were observed at the northernmost points (around 125-150 km upstream) of the August 2014, November 2013, and October 2014 cruises (Fig. 5).

**3.2 DIC and TA riverine flux**

We examined inputs of DIC and TA from the Delaware, Schuylkill, and Christina rivers from March to October 2015 (Fig. 2). As DIC and TA in the Delaware River and tributaries were only measured periodically, in order to estimate input fluxes precisely we first established a quantitative relationship between concentration and river discharge (Cai et al., 2008). We found that the observed DIC and TA concentrations in each tributary varied negatively with river discharge (Fig. 4). These relationships were used to estimate DIC and TA in the tributaries from average discharge measured for each cruise (Table 2), which were then combined with daily discharges recorded at each river from 2013 to 2015 to compute a more robust estimate. By this approach, we estimate that the annual flux of DIC and TA from the three rivers to the estuary was $11.0 \pm 5.8$ x $10^9$ and $10.8 \pm 5.1$ x $10^9$ mol C yr$^{-1}$, respectively.

**3.3 DIC and TA export flux**

DIC and TA values varied linearly with salinity near the ocean end-member value, suggesting no net addition or removal of DIC and TA. The effective river end-member concentrations of DIC and TA were calculated by extrapolating the DIC and TA conservative mixing lines from the high salinity waters to zero salinity (Fig. 3) (Cai et al., 2004; Guo et al., 2008). The difference between the effective and actual concentrations at the river end-member indicates the amount of DIC and TA added or removed during mixing and therefore not transported to the ocean (Boyle et al., 1974; Cai and Wang, 1998; Liu et al., 2014). Using the effective concentrations and the combined river discharge for the Delaware, Schuylkill, and Christina rivers recorded over the entire cruise period including discharges recorded 10 days prior to the survey, annual DIC and TA fluxes to the ocean were estimated to be $11.5 \pm 7.4 \times 10^9$ and $13.0 \pm 9.0 \times 10^9$ mol C yr$^{-1}$ (Table 3). Thus, DIC export flux out of the estuary is only 4.5% greater than the riverine flux into the estuary. However, TA flux increased 20% throughout the estuarine zone.

**4 Discussion**

**4.1 Influence of river discharge and weathering intensity**

The extensive and routine collection of water samples conducted by USGS allows us to explore long term trends in alkalinity (from the mid-20[th] to early 21[st] century) in the Delaware and Schuylkill rivers (USGS stations 01463500 and 01474500, respectively). For USGS alkalinity values, we use similar approaches as conducted in Stets et al., (2014). We combine 8 various parameter codes that include alkalinity, acid neutralizing capacity (ANC), or $HCO_3^-$ (Table 4). Alkalinity and ANC follow identical electrometric procedures except that alkalinity samples are filtered while ANC samples are not. The compilation of historical USGS water quality data from 1940 to the present shows that TA for the Delaware River at Trenton was negatively correlated with river discharge (Fig. 6). TA was highest during low flow season (fall) and lowest during high flow season (spring) (Fig. 6). Negative correlation between TA and river discharge has been observed for other river systems such as the Mississippi, Changjiang, Pearl, Huanghe, Congo, and Indus (Probst et al., 1992; Karim and Veizer, 2000; Cai, 2003; Li and Zhang, 2003; Chen et al., 2008; Guo et al., 2008; Liu et al., 2014). In the Delaware River, the highest TA fluxes occurred during peak flow season (spring) and the lowest TA fluxes occurred during the lowest flow season (fall) (Fig. 6). It is important to note that flux is governed by both river discharge and concentration. In the case of an extreme weather event, TA fluxes may be twice as large as the average flux. Under the same conditions, if river discharge is four-fold higher, concentrations must be reduced in half to yield a two-fold increase in TA flux. Thus, it appears that variation in TA (and DIC) is mainly a result of seasonal shifts in river discharge. Such fluctuations in river DIC and TA are expected as they are primarily governed by the dilution of weathering products by rain, and also are compensated by the increased weathering flux and other sources during wet seasons (White and Blum, 1995; White, 2003; Cai et al., 2008).

Another interesting but rarely reported phenomenon is the seasonal variation of the DIC to TA ratio at the freshwater end-members. At Trenton, the ratios (1.02 – 1.11) were highest during high discharge periods ($> 200$ m$^3$ s$^{-1}$) and lowest (0.86 – 1.01) at low discharge periods ($< 150$ m$^3$ s$^{-1}$) (Table 1). Similar results were found in the Schuylkill River where DIC to TA ratios were highest (1.02 – 1.07) at high discharge ($> 100$ m$^3$ s$^{-1}$) and lowest (0.93 to 1.02) during low discharge ($< 75$ m$^3$ s$^{-1}$) (Table 1). If only influenced by the weathering of carbonate and silicate minerals, the ratio of DIC to TA would remain close to unity (Cai et al., 2004). However, CO$_2$ production from soil

organic matter respiration and imbalances between production and respiration along the aquatic continuum can increase DIC to TA ratios (Mayorga et al., 2005). Presumably, during the wet season and high discharge periods, more $CO_2$ from soil organic matter respiration stored in the drainage basin is brought along the river system while less $CO_2$ is lost to the atmosphere due to a faster transport and lower surface area to volume ratio (i.e. deeper water depths) (Bass et al., 2014). We suggest that changes in the DIC to TA ratio at the freshwater end-member may reflect inputs of soil organic matter respiration due to seasonal variations in discharge, temperature, and moisture content, and less $CO_2$ degassing due to fast transport of water to the estuary. As the ratio of DIC to TA determines aquatic pH and the buffer capacity (Egleston et al., 2010), our observations indicate that variation of this ratio should be considered in future global carbon cycle models, in particular regarding how wet and drought cycles in future climate scenarios would affect coastal water acidification and how coastal waters will respond to a changing terrestrial carbon export (Reginer et al., 2013; Bauer et al., 2013).

**4.2 Influence of tributary mixing**

TA in the Schuylkill River was much higher than TA in the Delaware River near Philadelphia (Fig. 5). A compilation of historical data collected at two USGS stations in Philadelphia from 1940 to the present show that not only was alkalinity in the Schuylkill River negatively correlated with river discharge, but that during periods of low river discharge markedly high alkalinity was observed (Fig. 7A). Further, historical records agreed remarkably well with our alkalinity measurements. Over the past recent decades, after low river discharge ($< 100$ $m^3$ $s^{-1}$) alkalinity reached from 1300 to 2500 µmol $kg^{-1}$, nearly two-fold greater than alkalinity values observed the Delaware River end-member (Fig. 7B).

The mineralogy of the Schuylkill River drainage basin may have a significant effect on TA patterns throughout the Delaware estuarine system. Geographically, the lower Schuylkill drainage basin extends through the Piedmont province, underlain by a mixture of limestone, shale, gneiss, schist, and dolomite, before discharging into the Coastal Plain province and the Delaware River (Stamer et al., 1985). Within this region, the Schuylkill River flows through the Valley Creek basin in which 68% of the region is comprised of carbonate rocks (Sloto, 1990). The center of the basin, otherwise known as Chester Valley, is primarily underlain by easily eroded limestone and dolomite bedrock with regional flow discharging into the Schuylkill River. Thus, it is likely that high riverine TA in the Schuylkill River is due to the weathering of carbonate rocks in the lower

Schuylkill drainage basin. We contend that elevated DIC and TA values exhibited in the Delaware River near Philadelphia are the result of the mixing of relatively high carbonate freshwater from the Schuylkill River, specifically due to the chemical weathering of limestone and dolomite bedrock across the lower Piedmont province. It stands to reason that tributary contributions must be considered when addressing total riverine DIC and TA fluxes as differences in drainage basin mineralogy can have a substantial effect on the carbonate chemistry throughout regional watersheds.

**4.3 Historical trends in riverine alkalinity**

Over the past century, changes in land use activity have significantly impacted the watershed export of organic and inorganic carbon, acids, and nutrients to the coastal ocean (Duarte et al., 2013). Long-term USGS records of river alkalinity in the Schuylkill River show that not only are alkalinity and river discharge negatively correlated, but that over decadal periods alkalinity values have increased with time (Fig. 7A). Although changes were not as great, a similar increasing trend in river alkalinity was also observed in the historical USGS dataset for the Delaware River (Fig. 7B). A more comprehensive study by Kaushal et al. (2013) found that alkalinity increased at 62 of 97 rivers in the eastern U.S. over decadal time scales. Alkalinity did not significantly change at the remaining sites. Various factors can influence long-term trends in river alkalinity such as carbonate lithology, acid deposition, and topography in watersheds. Kaushal et al. (2013) suggested that increased acid deposition elevates riverine alkalinity by promoting weathering processes, particularly in watersheds with high carbonate lithology. Further, watershed elevation may be a good predictor for alkalization rates. Acid deposition may be greater at higher elevations, and such areas tend to have thinner soils and a weaker buffering capacity, increasing susceptibility to the effects of acid deposition. Recent studies show that human induced land-use changes such as deforestation, agricultural practices, and mining activities have direct impacts on the buffering capacity of streams and rivers (Brake et al., 2001; Oh and Raymond, 2006; Raymond and Oh, 2009). Through chemical weathering processes, enhanced precipitation and local runoff can also have huge effects on increased alkalinity in coastal ecosystems (Raymond et al., 2008). For example, it was suggested that over the past century, total alkalinity export from the Mississippi River to the Gulf of Mexico has risen by nearly 50% due to widespread cropland expansion and increased precipitation in the watershed (Raymond and Cole, 2003; Raymond et al., 2008). Similarly, Stets et al., (2014) explored historical time series of alkalinity values in 23 different

riverine systems throughout the U.S. They found that alkalinity increased at 14 of these locations mostly in the Northeastern, Midwestern, and Great Plains of the U.S. While alkalinity increased over time at most locations, it decreased in the Santa Ana, upper Colorado, and Brazos rivers. Factors contributing to decreasing alkalinity at these locations include dilution by water from external sources outside the basin and retention of weathering products in storage reservoirs.

While numerous studies indicate increasing alkalinity in estuarine waters, the impact of methodological changes over time cannot be neglected. Conveniently, USGS has published a series of manuals, both past and present, discussing the analytical procedures and methods followed during specialized work in water resources investigations (Woods, 1976; Fishman et al., 1989; Radke et al., 1998). Historically, the USGS measured alkalinity as fixed endpoint titrations on unfiltered samples, and commonly reported values as concentrations of bicarbonate (Clarke, 1924). By 1984, the USGS also began conducting fixed endpoint and incremental titrations on filtered samples (Raymond et al., 2009; Kaushal et al., 2013). Presently, USGS performs several variations of tests that describe the alkalinity, including standard alkalinity, acid neutralizing capacity, and carbonate alkalinity. Samples are measured using either a standard buret, micrometer buret, or by an automated digital titrator (Fishman et al., 1989; Radke et al., 1998). Micrometer burets offer higher accuracy and precision than standard burets while automated titrators are more preferred due to convenience and durability (Radke et al., 1998). Fixed endpoint titrations are generally less accurate than inflection point titrations, especially in low carbonate waters or areas with high organic and noncarbonated contributions to alkalinity (Radtke et al., 2008). Such methodological changes, however, would result in an underestimate of alkalinity if there is any (Kaushal et al., 2013). Thus, our conclusion of an increasing alkalinity trend in the Delaware River water will still hold and can be a conservative estimate. Such alkalinity increase has been observed throughout many river and estuarine systems (Raymond et al., 2003; Raymond et al., 2009; Duarte et al., 2013; Kaushal et al., 2013; Stets et al., 2014).

## 4.4 Seasonal variation in estuarine DIC

DIC in the Delaware Estuary also shifted with the seasons. In spring (March 2014 and April 2015) and summer (August 2014), DIC deviated slightly from conservative mixing in mid-salinity waters (S ~ 15 to 25) while TA varied linearly with salinity, suggesting consumption of $CO_2$ in the water column (Fig. 3). During the same time, pH was highest over the entire year, consistent with the presence of a phytoplankton bloom in spring and late summer (Fig. 3). Nonlinear distributions

were observed when plotting DIC against TA (Fig. 8). The curvature (concave upward trend at both ends) pattern indicates DIC removal in the mid-Delaware Bay during productive seasons. Joesoef et al., (2015) found that internal biological processes have significant effect on $CO_2$ dynamics within the Delaware Bay. In March and August 2014, $pCO_2$ was low ($160 - 350$ µatm)

and $CO_2$ uptake from the atmosphere was greatest ($-21 - 2.5$ mmol m$^{-2}$ d$^{-1}$) throughout the mid- and lower bay regions, indicating biological $CO_2$ removal (Joesoef et al., 2015). Thus, while not as large as changes in weathering and precipitation rates on DIC variability, internal biological processes within the bay system can lead to seasonal shifts in DIC concentrations.

Strong linear trends of TA with salinity across the estuarine mixing zone throughout all seasons suggest that the export of inorganic carbon from salt marshes to the main channel of the estuary is relatively small. If this was not the case, TA and DIC to salinity relationships would show a mid-point enrichment above the mixing line, as $SO_4^{2-}$ reduction is an important organic matter decomposition pathway that would generate $HCO_3^-$ in salt marshes (Cai and Wang, 1998; Jiang et al., 2008). Such humpback distribution is not observed. Nonetheless, it is evident that more research in estuarine systems is needed to accurately depict the influence of salt marsh exports on the carbonate chemistry of estuarine waters, especially in larger bay systems with long freshwater residence times.

**4.6 DIC mass balance**

Using freshwater discharge from the Delaware, Schuylkill, and Christina rivers (Table 1), DIC input fluxes to the estuary were computed for each cruise based on the linear relationships shown in Fig. 4. Combining total DIC fluxes for each river, we obtain an annual-averaged DIC input flux of $11.0 \pm 5.8$ x $10^9$ mol C yr$^{-1}$. Using the effective concentrations extrapolated from the high salinity water, an annual-averaged DIC export flux to the ocean of $11.5 \pm 7.4 \times 10^9$ mol C yr$^{-1}$ was calculated. Since approximately 70% of the freshwater input to the estuary comes from the Delaware, Schuylkill, and Christina rivers, and the remaining percentage comes from small rivers, nonpoint source runoff, and waste water treatment facilities, we estimate that the Delaware, Schuylkill, and Christina rivers provide the estuary with about 70% of its total freshwater input, calculated from the combined annual mean discharge (387 m$^3$ s$^{-1}$) of these rivers from 2013-2015. By upward scaling, we obtain an annual mean discharge of 553 m$^3$ s$^{-1}$ and a final DIC input flux of $15.7 \pm 8.2 \times 10^9$ mol C yr$^{-1}$ and export flux of $16.5 \pm 10.6 \times 10^9$ mol C yr$^{-1}$. We acknowledge that average riverine DIC and TA concentrations from remaining small rivers and nonpoint source

runoff are not necessarily equivalent to the weighted DIC and TA averages for the Delaware, Schuylkill, and Christina rivers. As such uncertainties are most often neglected, it is necessary to consider their effect on final flux estimates. However, since additional research and data collection are needed, here we assume that the mineralogy and drainage basins of the remaining 30% yield similar carbonate concentrations as Delaware's three major river systems. In this study, we upscaled both the river-to-estuary flux and the estuary-to-offshore flux by the same proportion (10/7) to estimate the total estuarine input and export fluxes. Thus, the uncertainty derived from upscaling would partially cancelled out and not substantially affect the conclusions discussed below. Annual air-water $CO_2$ flux to the atmosphere from the Delaware Estuary has recently been estimated as $2.4 \pm 4.8$ mol C m$^{-2}$ yr$^{-1}$ (Joesoef et al., 2015). Using the annual air-water $CO_2$ flux and an estimated surface water area of 1773 km$^2$ for the estuarine system (Sutton et al., 1996), the total $CO_2$ flux to the air is estimated as $4.3 \times 10^9$ mol C yr$^{-1}$. Thus, a speculative DIC mass balance for the estuary is as follows:

River input flux ($15.7 \times 10^9$ mol C yr$^{-1}$)

+ Internal estuarine $CO_2$ production (?)

+ Inputs from surrounding salt marshes (?)

+ Inputs from benthic recycling (?)

= Estuarine output flux ($16.5 \times 10^9$ mol C yr$^{-1}$)

+ Atmospheric flux ($4.3 \times 10^9$ mol C yr$^{-1}$)

The sum of the unknown internal DIC production terms is estimated at $5.1 \times 10^9$ mol C yr$^{-1}$. This internal DIC production includes respiration in the water column and benthos, $CO_2$ addition from intertidal marsh waters, wastewater effluents, ground water discharge, and other various external sources. If we pool water column and benthic respiration into one term and ignore additional input from wastewater effluents and ground water discharge, DIC fluxes can be viewed as a measure of net ecosystem production (NEP). Using DIC input and export fluxes and air-water $CO_2$ fluxes from Joesoef et al. (2015), we estimate NEP during each cruise as described above (Fig. 9). In early spring, positive NEP indicates that the estuary is net autotrophic ($10.3 \pm 2.0$ mmol C m$^{-2}$ d$^{-1}$), and exports or stores an excess of organic carbon. A shift to negative NEP in the summer (-9.8 $\pm$ 11.6 mmol C m$^{-2}$ d$^{-1}$) indicates a net heterotrophic system where ecosystem metabolism is

sustained by external inputs of organic matter (Fig. 9). In contrast, from fall to early winter season, the estuary fluctuates from a near balanced ecosystem to a net heterotrophic environment.

Other studies have explored NEP across the estuarine gradient of the Delaware Estuary (Sharp et al., 1982; Lipschultz et al., 1986; Hoch and Kirchman, 1993; Preen and Kirchman, 2004).

Significant depletion of dissolved oxygen and supersaturation of $pCO_2$ levels in freshwaters (salinity $<$ 10), suggests that the upper estuary is heterotrophic while the lower estuary is autotrophic (Sharp et al., 1982). More recent studies have found that respiration often exceeds primary production in the upper Delaware River (Hoch and Kirchman, 1993; Preen and Kirchman, 2004). Comparably, Culberson (1988) used inorganic carbon and dissolved oxygen measurements

to estimate apparent carbon production and oxygen utilization throughout the Delaware Estuary. Similar to our spring NEP results, Culberson (1988) found that during the months of March to May from 1978 to 1985, most of the estuary ($6 < S < 30$) suffered a net inorganic carbon loss. Presumably, this loss occurred during the spring phytoplankton bloom, a period of intense inorganic carbon uptake by phytoplankton. While respiration rates often outweigh primary

production in the upper tidal river, generally net community production increases down the estuary, transitioning to a near balanced to autotrophic system in the mid- to lower bay regions (Hoch and Kirchman, 1993; Preen and Kirchman, 2004).

Despite high $CO_2$ consumption during the spring and late summer, annually the Delaware Estuary is a weak source of DIC with an NEP = -1.3 $\pm$ 3.8 mol C m$^{-2}$ yr$^{-1}$, which is in sharp contrast to

many smaller river estuaries that exhibit strong net heterotrophy (-17 $\pm$ 23 mol C m$^{-2}$ yr$^{-1}$) (Borges and Abril, 2011). Of the 79 estuarine studies compiled by Borges and Abril (2011) that reported gross primary production (GPP), community respiration (CR), and NEP rates, overall only 12 estuaries are net autotrophic. Most estuaries are strongly net heterotrophic probably because of high inputs of labile organic matter from tributaries that support CR while GPP is reduced due to

limited light availability caused by elevated suspended matter (Smith and Hollibaugh, 1993; Heip et al., 1995; Gattuso et al., 1998; Gazeau et al., 2004; Borges and Abril, 2011). However, the relationship between NEP and GPP varies considerably across different estuaries depending on factors such as the degree of light limitation, the fraction of inorganic nutrient to organic carbon inputs, and the size of the estuarine system, with smaller estuaries showing increased heterotrophy

over larger systems such as the Delaware Bay (Hopkinson, 1988; Heip et al., 1995; Kemp et al., 1997; Caffrey, 2004; Borges and Abril, 2011).

Riverine input and estuarine export fluxes varied greatly over time and are largely governed by seasonal discharge patterns (Table 2 and 3). The highest fluxes occurred during spring when discharge was high while the lowest values occurred in the fall and winter when discharge was low. However, seasonal changes in NEP did not reflect variations in river discharge. Discharge decreased throughout the year while NEP rates fluctuated across seasons (Fig. 9). Instead, NEP largely mirrored seasonal variations in air-water $CO_2$ fluxes. When the estuary acted as a source of $CO_2$, NEP was negative while when the system was a $CO_2$ sink, NEP was positive. From the annual mass balance model, the small difference between riverine input and export flux suggests that the majority of DIC produced within the estuary is exchanged with the atmosphere rather than exported to the ocean. More research and data are needed to accurately ascertain seasonal variations in estuarine fluxes and NEP.

Unlike in most previously studied estuaries, but similar to the macro-tidal Scheldt Estuary, freshwater residence time in the Delaware Bay is generally long ranging from about one to a few months (Gay and O'Donnell, 2009; Borges and Abril, 2011). In contrast, the smaller stratified Randers Fjord has a much shorter residence time (few days) (Nielsen et al., 2001). In the smaller Randers Fjord, $CO_2$ emission to the atmosphere is lower than net community production (NCP) in the mixed layer or much less significant (Gazeau et al., 2005). This occurrence is partly due to the decoupling in ecosystem production caused by water stratification. As organic matter is produced in the surface waters, its degradation occurs in the bottom waters, and ultimately delaying $CO_2$ exchange with the atmosphere (Borges and Abril, 2011). Further, total DIC export to the Baltic Sea is higher than riverine DIC inputs to the Randers Fjord, suggesting that, due to the shorter freshwater residence times of systems, much of the DIC produced by net respiration is exported rather than removed to the atmosphere (Gazeau et al., 2005). Comparably, the Rhine exhibits extremely short freshwater residence time (~2 days) due to intense freshwater discharge (~2200 $m^3 \, s^{-1}$). Such rapid turnover time, leads to reduced emission of methane ($CH_4$) to the atmosphere by bacterial oxidation and smaller internal DIC production due to net heterotrophy (Borges and Abril, 2011). A similar case study was seen for the rapidly transiting Altamaha River in the U.S. southeastern margin (Cai and Wang 1998; Jiang et al., 2008). However, lateral inputs from intertidal marsh systems in small estuaries can enhance accumulation and degradation of organic matter in surface waters, resulting in high $CO_2$ degassing fluxes (Dai and Wiegert, 1996; Cai and Wang, 1998; Neubauer and Anderson, 2003).

Due to the large size of the Delaware Bay, the effect from the production and decomposition of marsh plants on $CO_2$ flux dynamics in the system may not be as influential as in smaller estuaries except near the coastlines where tides regularly flush marsh boundaries (Joesoef et al., 2015). In this study, we did not sample the sub-estuaries within nor areas near the perimeters of the bay, but instead were limited to sampling within the main channel of the estuary. We note while the Delaware River is only a medium size river, the Delaware Bay is one of the largest bays in the U.S. eastern coast and its hydrodynamics is largely controlled by the exchange with the ocean (residence time of 1-3 months). In the Scheldt Estuary, long freshwater residence time typically leads to DIC accumulation in the water column (Abril et al., 2000; Borges et al., 2006). In addition, in both the Delaware and Scheldt estuaries, small differences between riverine input and export flux suggests that the majority of DIC produced within the estuary is exchanged with the atmosphere rather than exported to the ocean. While similar NEP values may be observed, the enrichment of DIC in estuarine waters and resulting $CO_2$ exchange with the atmosphere will be more intense in estuarine systems with long residence times versus estuaries with short residence times (Borges and Abril, 2011). Thus, we suspect that in estuaries with long freshwater residence times (i.e. the Delaware Estuary), much of the DIC produced by NEP is most likely removed to the atmosphere rather than exported to the sea.

**5 Conclusion**

Strong negative correlations between river TA and freshwater discharge in the non-tidal Delaware, Schuylkill, and Christina rivers suggest that changes in $HCO_3^-$ concentrations in the Delaware Estuary reflect dilution of weathering products in the drainage basin. Elevated DIC and TA concentrations near the Philadelphia region in the upper estuary are largely the result of relatively high carbonate freshwater from the lower Schuylkill River drainage basin, a consequence of chemical weathering of limestone and dolomite bedrock. Increased alkalinity in the Delaware and Schuylkill rivers over the past 70 years coincide with global trends toward higher alkalinity in river and estuarine waters over decadal timescales. In addition to strong variations in discharge and mixing from the three rivers, seasonal changes in NEP within the estuary also contribute to shifts in DIC concentrations. Lastly, a preliminary mass balance analysis indicates only a small difference between riverine DIC input and export flux suggesting that in the Delaware Estuary and other estuarine systems with long freshwater residence times that much of the DIC produced by

NEP or supplied from surrounding marshes is most likely emitted to the atmosphere rather than exported to the sea.

## 6 Acknowledgements

Cai acknowledges UD internal funds and the National Aeronautics and Space Administration (NNX14AM37G) for supporting his research. DLK was supported by NSF OCE-1030306 and OCE-1261359. The cruises were supported by awards from the National Science Foundation (OCE-1155385, OCE-1261359, and OCE-1030306), the Delaware Sea Grant College Program (RHCE14-DESG). In addition, we would like to thank the captains and crew of the R/V Hugh R. Sharp and the R/V Joanne Daiber for their tremendous support. We also thank J. H. Sharp, G. W. Luther III, J. H. Cohen, and B. J. Campbell for providing us the opportunity to participate on their research cruises.

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

**Table 1. Sampling dates, average discharge, pH, DIC, TA, and DIC to TA ratio in the Delaware (Trenton), Schuylkill, and Christina rivers.**

| Location | Date | Discharge ($m^3 s^{-1}$) | $pH_{NBS}$ (at 25°C) | DIC ($\mu mol\ kg^{-1}$) | TA ($\mu mol\ kg^{-1}$) | DIC:TA |
|---|---|---|---|---|---|---|
| Trenton | 3/10/2015 | 182 | 8.8 | 973.4 | 1038.8 | 0.94 |
| | 4/21/2015 | 442 | 7.8 | 745.2 | 723.7 | 1.03 |
| | 5/7/2015 | 190 | 8.8 | 856.5 | 902.9 | 0.95 |
| | 5/21/2015 | 148 | 8.0 | 1025.5 | 1015.9 | 1.01 |
| | 6/9/2015 | 199 | 8.2 | 857.8 | 869.2 | 0.99 |
| | 6/23/2015 | 425 | 7.7 | 783.5 | 765.5 | 1.02 |
| | 7/2/2015 | 1127 | 7.2 | 454.2 | 410.4 | 1.11 |
| | 9/15/2015 | 183 | 8.2 | 945.8 | 936.7 | 1.01 |
| | 9/29/2015 | 98 | 8.7 | 945.8 | 1103.9 | 0.86 |
| | 10/12/2015 | 170 | 8.1 | 1095.2 | 1046.1 | 1.05 |
| Schuylkill | 4/16/2015 | 52 | 8.9 | 1421.2 | 1525.7 | 0.93 |
| | 5/21/2015 | 32 | 8.1 | 1682.9 | 1655.9 | 1.02 |
| | 6/9/2015 | 105 | 7.9 | 1400.1 | 1371.3 | 1.02 |
| | 7/2/2015 | 271 | 7.7 | 1095.3 | 1026.3 | 1.07 |
| | 9/15/2015 | 60 | 7.8 | 1506.1 | 1472.2 | 1.02 |
| | 9/29/2015 | 19 | 8.1 | 2071.3 | 2107.8 | 0.98 |
| | 10/12/2015 | 34 | 8.3 | 1869.3 | 1851.4 | 1.01 |
| Christina | 4/16/2015 | 14 | 7.7 | 1056.5 | 1015.1 | 1.04 |
| | 4/28/2015 | 15 | 7.5 | 1076.4 | 1018.6 | 1.06 |
| | 5/21/2015 | 11 | 7.7 | 1134.1 | 1072.8 | 1.06 |
| | 6/9/2015 | 32 | 7.5 | 1089.4 | 1004.0 | 1.08 |
| | 9/15/2015 | 6 | 7.9 | 1326.9 | 1210.6 | 1.10 |
| | 9/29/2015 | 7 | 8.0 | 1188.6 | 1165.4 | 1.02 |
| | 10/12/2015 | 7 | 8.0 | 1199.6 | 1168.0 | 1.03 |

**Table 2. Estimated TA and DIC in the Delaware (Trenton), Schuylkill, and Christina rivers, calculated by linear regression using discharge and their input fluxes to the Delaware Estuary.**

| Survey | Trenton | | Schuylkill | | Christina | | TA | DIC |
| | TA | DIC | TA | DIC | TA | DIC | input flux | input flux |
| | ($\mu$mol kg$^{-1}$) | ($\mu$mol kg$^{-1}$) | ($\mu$mol kg$^{-1}$) | ($\mu$mol kg$^{-1}$) | ($\mu$mol kg$^{-1}$) | ($\mu$mol kg$^{-1}$) | ($10^9$ mol yr$^{-1}$) | ($10^9$ mol yr$^{-1}$) |
|---|---|---|---|---|---|---|---|---|
| March 2014 | 700.3 | 721.5 | 1341.8 | 1366.8 | 935.3 | 1004.8 | 15.6 | 16.0 |
| April 2015 | 609.1 | 647.6 | 1382.8 | 1404.3 | 1030.1 | 1093.7 | 16.7 | 17.6 |
| June 2013 | 634.0 | 667.8 | 995.4 | 1050.5 | 870.7 | 944.2 | 21.3 | 22.5 |
| July 2014 | 901.6 | 884.7 | 1565.3 | 1571.0 | 1050.7 | 1113.0 | 9.7 | 9.6 |
| August 2014 | 1101.0 | 1046.4 | 1977.9 | 1947.8 | 1131.8 | 1188.9 | 5.4 | 5.2 |
| October 2014 | 1147.2 | 1083.9 | 1860.7 | 1840.8 | 1123.1 | 1180.8 | 5.3 | 5.1 |
| November 2013 | 1154.0 | 1089.4 | 1929.7 | 1903.7 | 1112.0 | 1170.3 | 5.0 | 4.8 |
| December 2014 | 998.9 | 963.7 | 1548.0 | 1555.1 | 1057.9 | 1119.7 | 8.4 | 8.3 |
| Annual Average | 894.6 | 879.1 | 1568.2 | 1573.6 | 1044.4 | 1107.0 | 10.8 | 11.0 |

**Table 3: Effective TA and DIC as a function of salinity, calculated by linear regression using data from high salinity waters in the Delaware Estuary and their export fluxes to the ocean.**

| Survey | Effective TA ($\mu$mol kg$^{-1}$) | | | Effective DIC ($\mu$mol kg$^{-1}$) | | | Total discharge (m$^3$ s$^{-1}$) | TA export flux ($10^9$ mol yr$^{-1}$) | DIC export flux ($10^9$ mol yr$^{-1}$) |
|---|---|---|---|---|---|---|---|---|---|
| | Slope | Intercept | $R^2$ | Slope | Intercept | $R^2$ | | | |
| Mar 2014 | 35.99 | 1034 | 0.97 | 35.59 | 889 | 0.97 | 597 | 19.5 | 16.7 |
| Apr 2015 | 37.33 | 1071 | 0.99 | 40.73 | 714 | 0.99 | 740 | 25.0 | 16.7 |
| Jun 2013 | 37.91 | 978 | 0.93 | 32.03 | 948 | 0.94 | 895 | 27.6 | 26.8 |
| Jul 2014 | 51.05 | 532 | 0.96 | 46.21 | 514 | 0.90 | 297 | 5.0 | 4.8 |
| Aug 2014 | 36.63 | 974 | 0.97 | 37.99 | 747 | 0.94 | 139 | 4.3 | 3.3 |
| Oct 2014 | 37.45 | 954 | 0.98 | 28.69 | 1087 | 0.97 | 129 | 3.9 | 4.4 |
| Nov 2013 | 28.48 | 1261 | 0.99 | 20.27 | 1360 | 0.98 | 124 | 4.9 | 5.3 |
| Dec 2014 | 35.28 | 1119 | 0.99 | 25.16 | 1219 | 0.96 | 234 | 8.3 | 9.0 |
| Annual Average | | | | | | | 387 | 13.0 | 11.5 |

**Table 4. USGS parameter codes used during analysis.**

| Parameter Code | Parameter Description | Total Count | Percentage of Total Count |
|---|---|---|---|
| 00410 | Acid neutralizing capacity, water, unfiltered, fixed endpoint titration, field | 920 | 28.5 |
| 00419 | Acid neutralizing capacity, water, unfiltered, inflection-point titration, field | 25 | 0.8 |
| 00440 | Bicarbonate, water, unfiltered, fixed endpoint titration, field | 1529 | 47.4 |
| 00450 | Bicarbonate, water, unfiltered, inflection-point titration, field | 25 | 0.8 |
| 00453 | Bicarbonate, water, filtered, inflection-point titration, field | 86 | 2.7 |
| 29801 | Alkalinity, water, filtered, fixed endpoint titration, laboratory | 133 | 4.1 |
| 39086 | Alkalinity, water, filtered, inflection-point titration, field | 283 | 8.8 |
| 90410 | Acid neutralizing capacity, water, unfiltered, fixed endpoint titration, laboratory | 224 | 6.9 |

**Figure captions**

**Figure 1. Map of the Delaware Estuary and river tributaries. Gray stars indicate USGS gauging stations (1) 01463500, (2) 01474010, (3) 01474500, (4) 01481500, (5) 01480015, (6) 01479000, and (7) 01478000. Black arrows indicate river names.**

**Figure 2. Daily discharge at the Delaware (Trenton), Schuylkill, and Christina rivers from March to October 2015. Note the different scales used for each river. Red diamonds indicate exact sampling dates. Green lines are when river waters were frozen.**

**Figure 3. Salinity distributions of DIC, alkalinity, and pH in the Delaware Estuary.**

**Figure 4. Alkalinity and DIC versus log discharge at the Delaware (Trenton), Schuylkill, and Christina rivers. Note the different scales used for each river.**

**Figure 5. Spatial distribution of alkalinity, DIC, and pH in the Delaware Estuary from the mouth of the bay (0 km) to the head of the tide at Trenton, NJ (215 km).**

**Figure 6. Relationship between alkalinity and Delaware River discharge at Trenton (1940 – 2015) (top). Black circles indicate data obtained from the USGS station while red circles indicate data collected in this study. Seasonal river discharge versus alkalinity and alkalinity flux for the same time period (bottom). Errors bars represent one standard deviation of the mean value for each month.**

**Figure 7. (a) Time series of the Schuylkill River discharge at Philadelphia, PA and (b) the Delaware River discharge at Trenton, NJ against alkalinity from 1940 to 2016. Note the different scales used for each river.**

**Figure 8. DIC versus alkalinity measured along the axis of the Delaware Estuary.**

**Figure 9. Seasonal variations of net ecosystem production, air-water $CO_2$ fluxes, and discharge in the Delaware Estuary. Note the different scales used for each plot. Discharge is defined as the average of the total discharge to the estuary recorded during each cruise period including discharges recorded 10 days prior to the survey.**

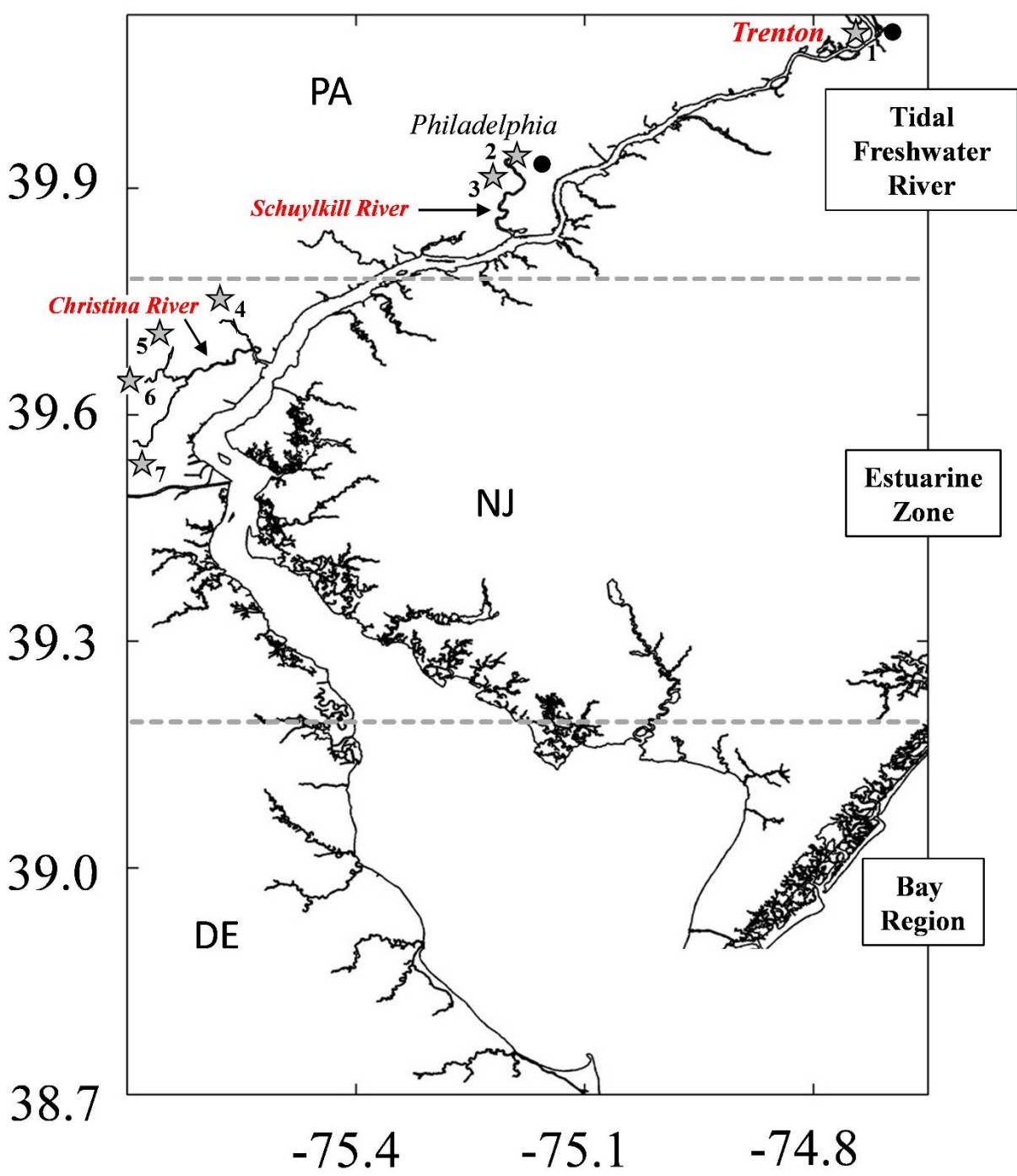

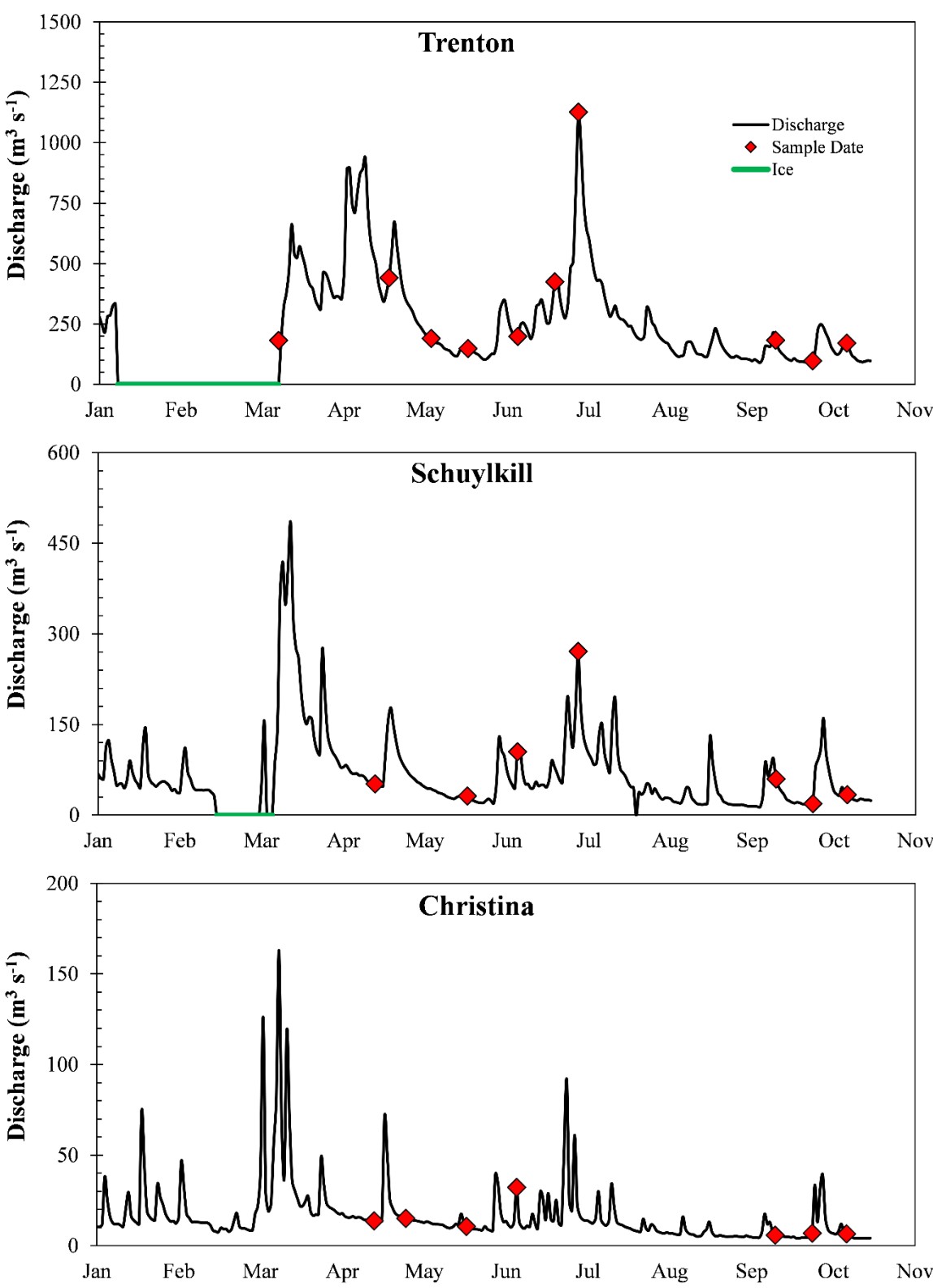

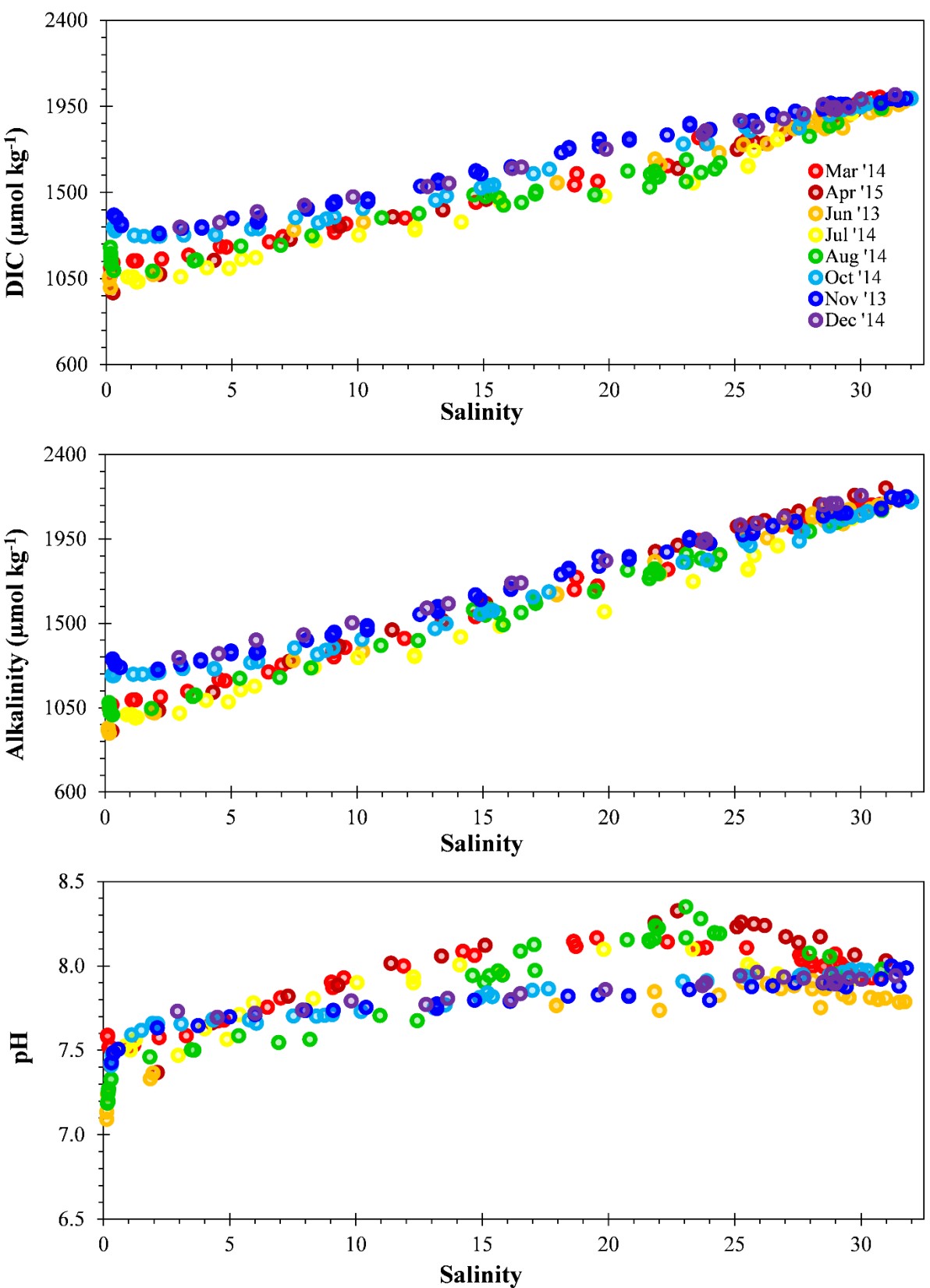

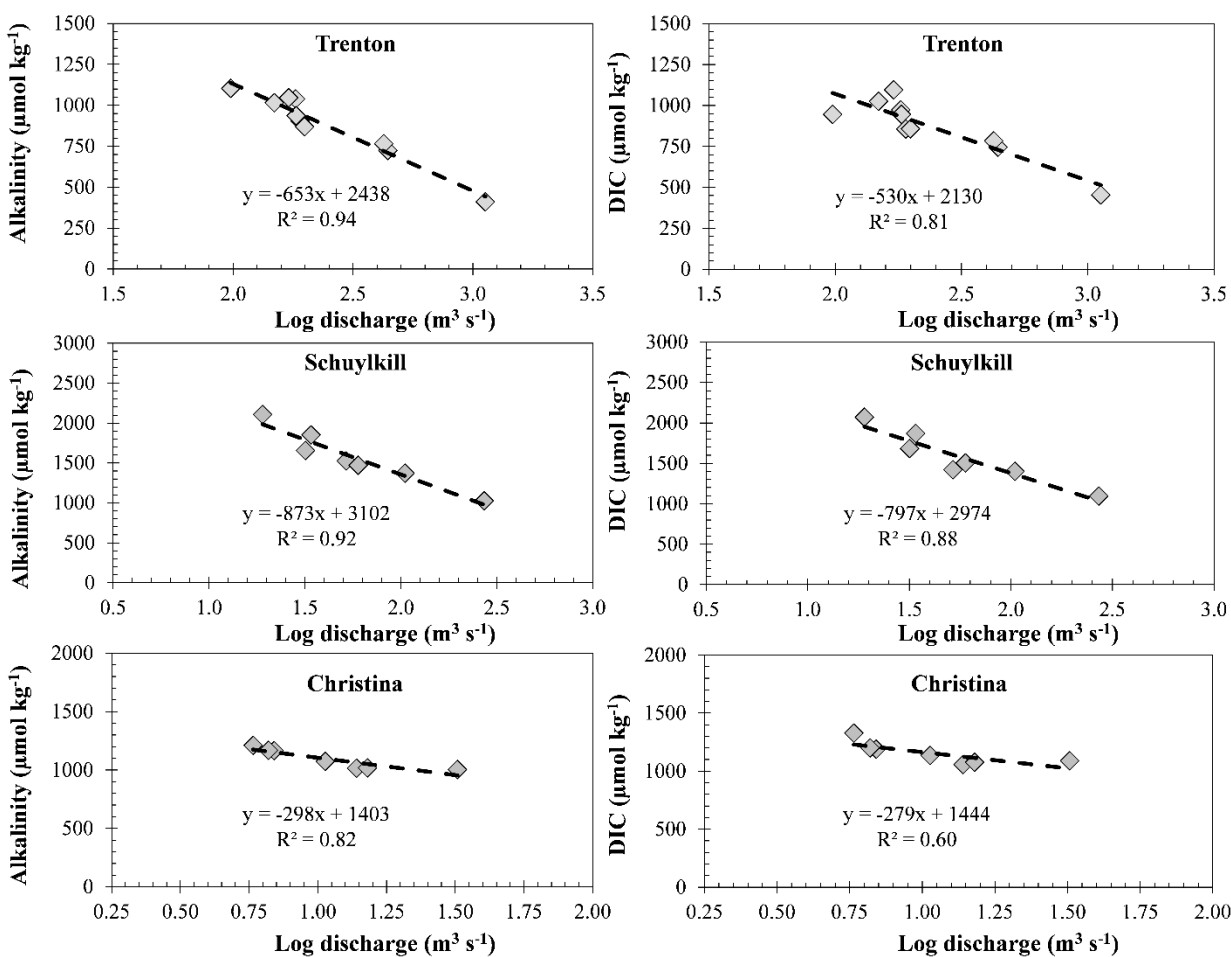

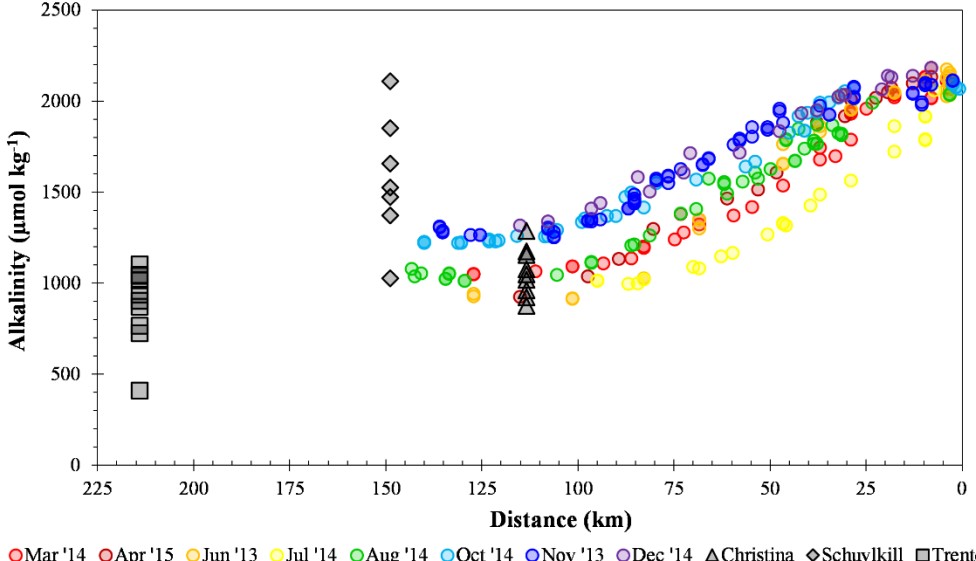

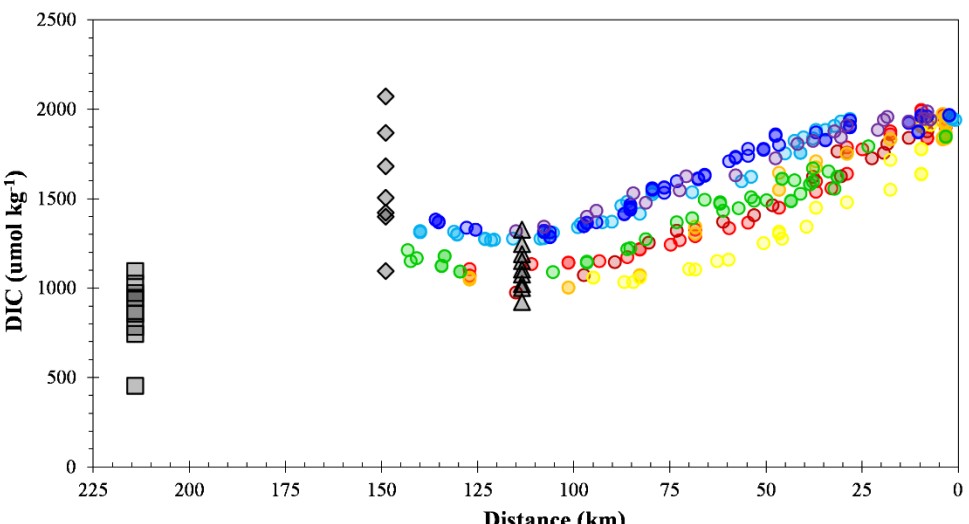

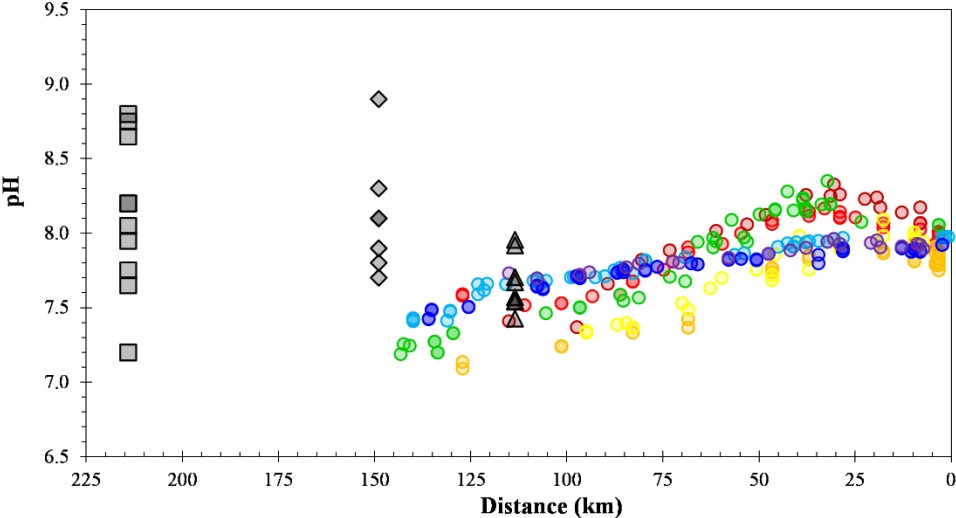

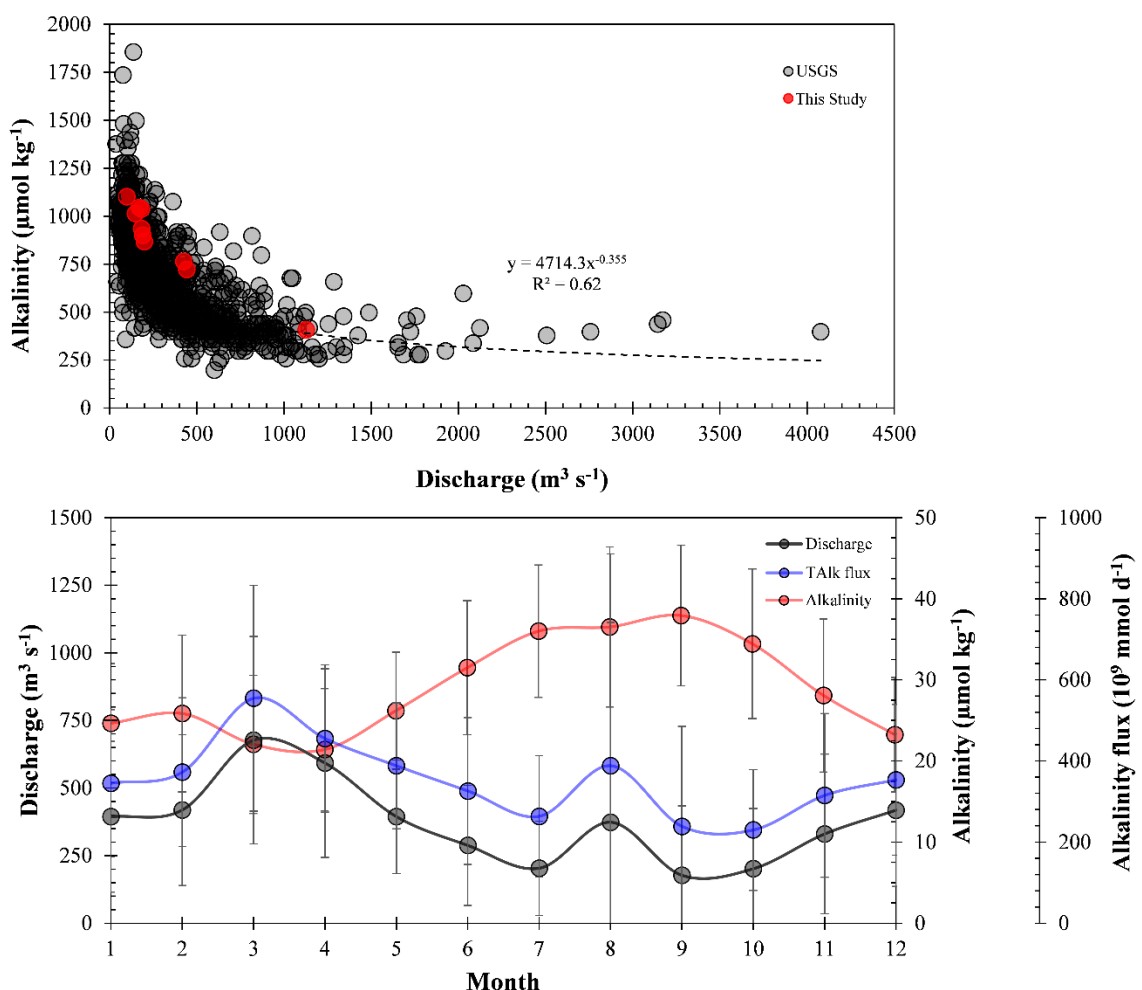

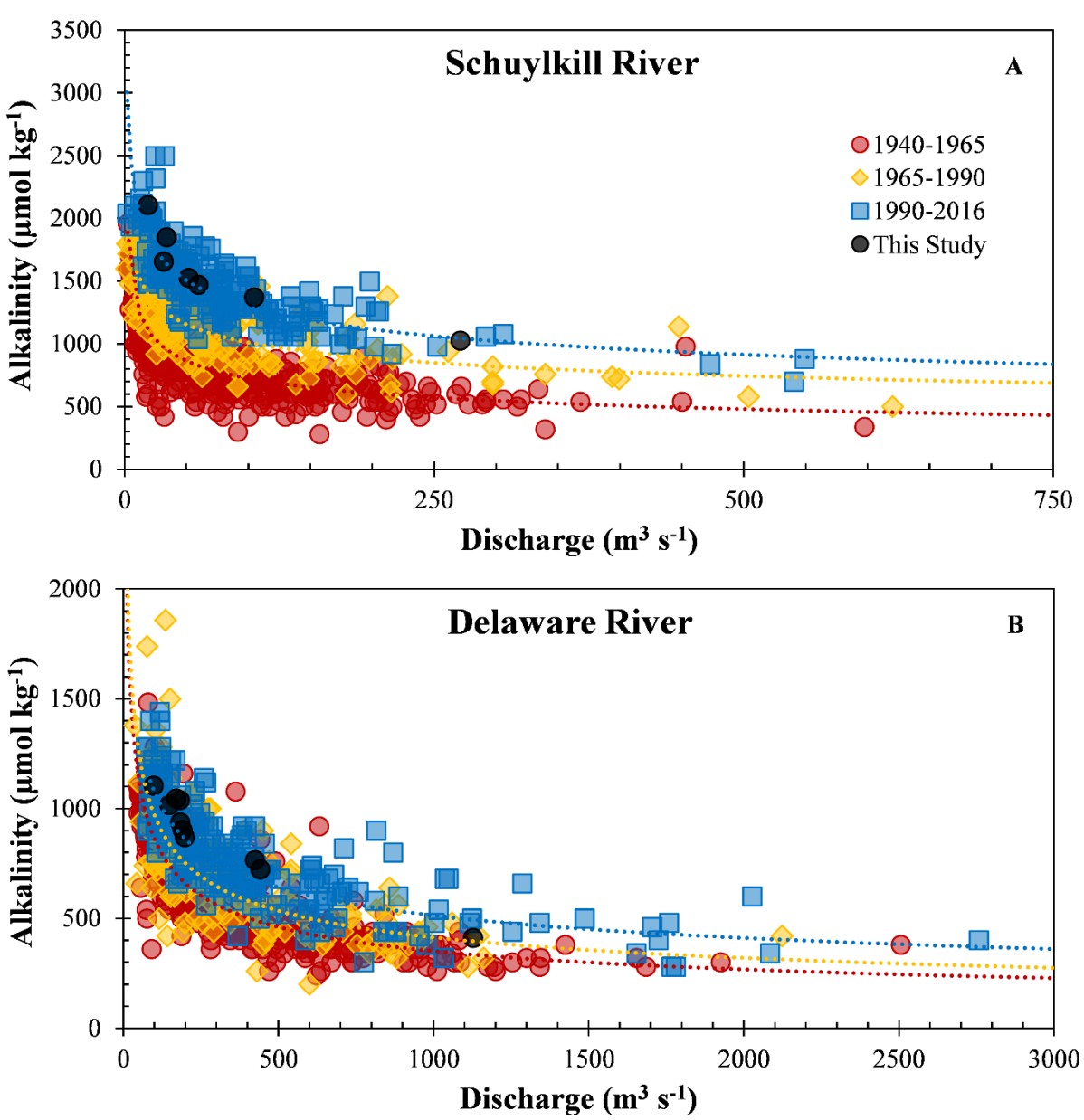

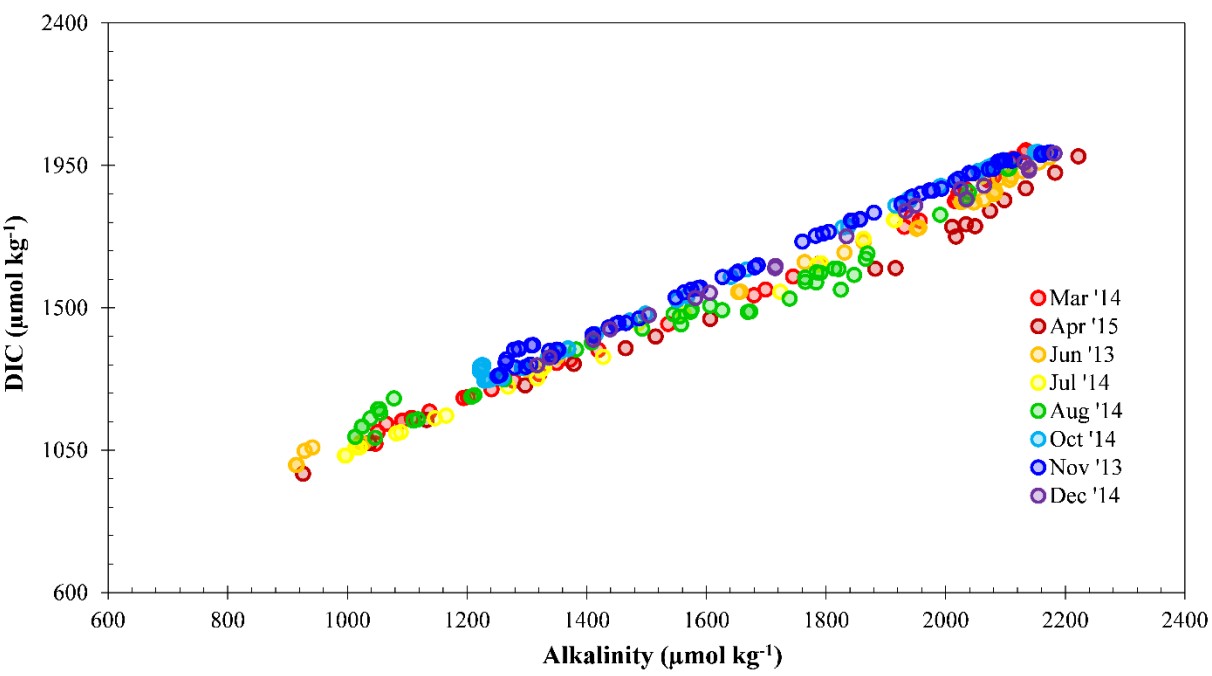

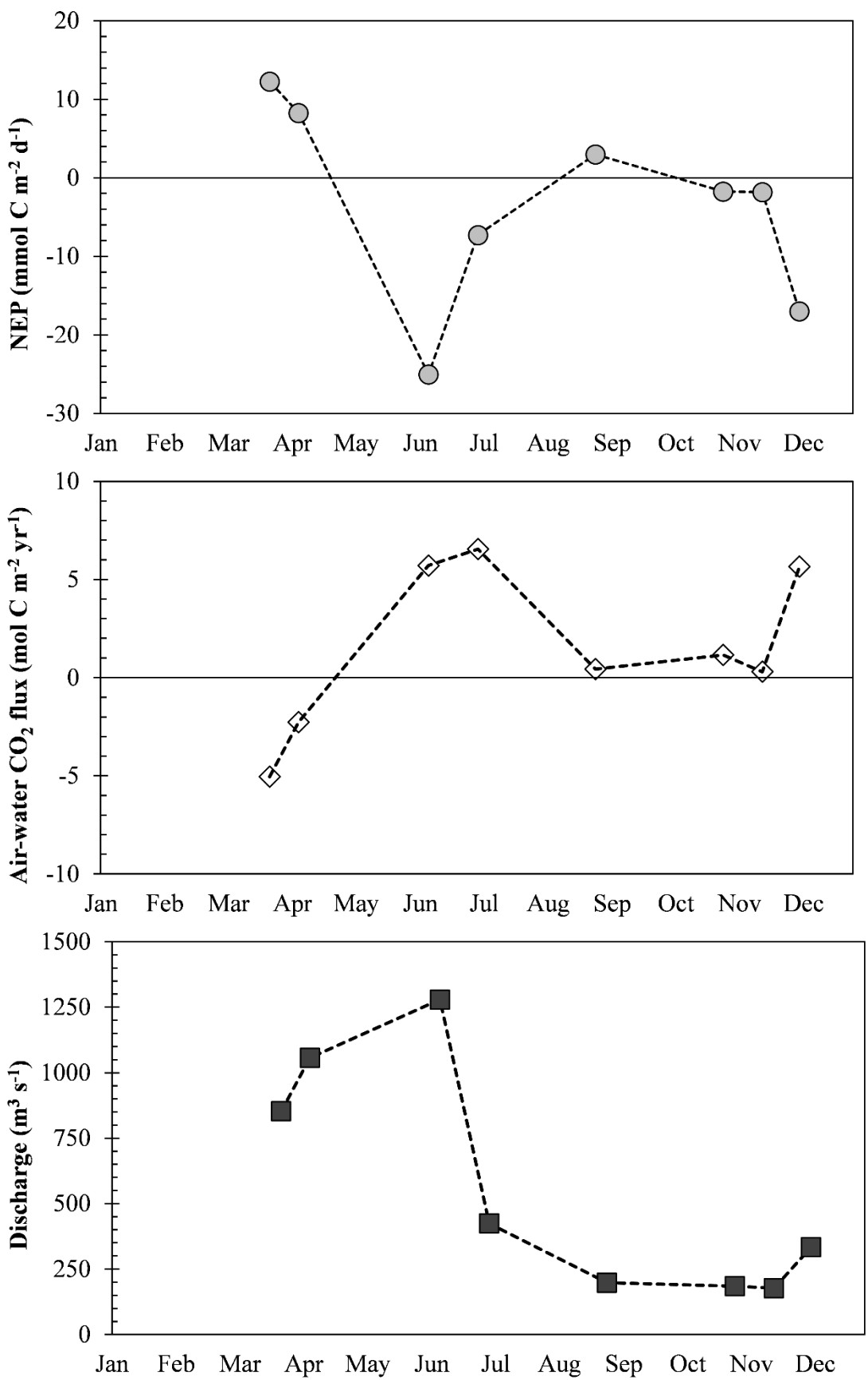