# Peer review of "Seasonal variability of the inorganic carbon system in a large coastal plain estuary"

_Biogeosciences, 2017_

## Referee Comment (RC1) · Anonymous Referee #1 · 4 Jul 2017

GENERAL COMMENTS

The authors report a high-quality data-set of total alkalinity (TA) and dissolved inorganic carbon (DIC) in the Delaware estuary. The paper is well written and was prepared with care, although there is nothing radically new, confirming previous findings (long-term trends in river alkalinity, residence time explains differences in carbon cycling from one estuary to another, . . .).

SPECIFIC COMMENTS

I suggest that the authors make their data-set publically available as a supplement of the paper.

Part of the paper is based on the analysis of long-term data-sets from the USGS, going

back to the 1940's. The authors compare their own data with the recent USGS (Figure 7), which validates the quality of the recent USGS data. But this does not necessarily mean that the old data are of the same quality, meaning the derived trends over the decades could be methodological. Please add in the discussion, some elements on the methods of TA analysis, data quality check, and any other element that might be useful to show that over the last 70 years the USGS data-set is of uniform quality and that the observed changes are real rather than methodological.

P2 L 16: to the list of processes that control CO2 in rivers, you could mention inputs from wetlands (Abril et al. 2014).

P3 L 10: I assume that this statement is based on some sort of analysis of numbers, so could you please state the range and central value of the area and the residence time of the estuaries from the cited studies.

P3 L 11: Please define the criteria (threshold value ?) and quantity (surface area ? discharge ? drainage areas ? Length ?) to distinguish "large" and "small" estuaries.

P7 L 12-13: The correlation between TA fluxes and discharge is due to auto-correlation. If you plot AxB versus B, you'll always generate a good correlation (Berges 1997), specially if B changes over several orders of magnitude (unlike A).

P7 L 30: it should be noted that the Amazon river has a very low TA, and extremely organic rich soils. This interpretation might not be applicable to a high TA river, with moderately organic rich soils such as the Delaware. Also, there is a very strong hydrological connectivity between the Amazon and soils through inundation of floodplains that is probably not as extensive in the Delaware.

P 10 L 25: The finding that intertidal marshes have little influence on the CO2 dynamics of the Delaware is quite interesting and would contradict the main conclusion and among the opening statements of the Cai (2011) paper: "It is demonstrated here that CO2 release in estuaries is largely supported by microbial decomposition of highly

productive intertidal marsh biomass".

P 13 L 6: This discussion seems to contradict the Introduction (P3 L10) that previously studied estuaries have a "short residence time"

Figure 1: axis legends have a different font from all of the other figures, it is advisable to have a uniform font in all figures.

Figure 4: Two decimals for $R^2$ are sufficient.

In some figures, numbers in axis legend have thousands separated by comma, but not in others. It is advisable to make this uniform.

In some figures, the axis name is "alkalinity", in others it is "TA". It is advisable to make this uniform

Figures 6 and 7: It is odd that one of the data-sets is named after one of the authors ("Cai"), I suggest that the data set is named "this study", something neutral and a bit more modest.

REFERENCES

Abril, G., Martinez, J.-M., Artigas, L.F., Moreira-Turcq, P., Benedetti, M.F., Vidal, L., Meziane, T., Kim, J.-H., Bernardes, M.C., Savoye, N., Deborde, J., Albéric, P., Souza, M.F.L., Souza, E.L., Roland, F., 2014. Amazon River Carbon Dioxide Outgassing fuelled by Wetlands. Nature 505, 395-398.

Berges J.A., 1997. Ratios, regression statistics, and "spurious" correlations, Limnol. Oceanogr. 42(5), 1006-1007

Cai W.-J., 2011. Estuarine and Coastal Ocean Carbon Paradox: CO2 Sinks or Sites of Terrestrial Carbon Incineration? Annu. Rev. Mar. Sci. 3, 123-45

---

## Referee Comment (RC2) · Anonymous Referee #2 · 11 Jul 2017

General comments

This manuscript deals with the seasonal variability of the carbonate system in the Delaware estuary, a large estuary along the eastern U.S. coast, and focuses specifically at quantifying the role of riverine DIC and TA inputs therein. It presents high-quality data obtained by the authors in both the estuary and rivers, as well as daily discharge and long-term riverine monitoring data from USGS. I overall enjoyed reading the manuscript and consider it robust, but I don't find its main conclusions very remarkable and was somehow left with the feeling that more could be done with the available data.

The authors mention studying the DIC:TA ratio as a relative novelty in this manuscript (I find the term "rarely studied" as used in the abstract a bit strong) and I agree that

this should deserve more attention in the literature in the context of minimum buffering capacity at the point where DIC≈TA. However, when presenting these data and trying to put so much focus on them, I expected the authors to do some more quantitative work on this, e.g. how the position of the DIC=TA point in the estuary varied over the course of the year and what role the riverine input played therein. This would, in my opinion, increase the impact of this work.

The manuscript seems to be written by multiple authors coming from different background (freshwater versus marine communities). This leads to overall imbalances in the manuscript, with certain topics being discussed much more extensively, and also more quantitatively, than others. Specific examples are given below. The authors should generally try to better unite the several parts of the manuscript.

Specific comments

I found the introduction particularly unbalanced. Specifically, I think that the first paragraph of the introduction (p. 2, l. 7-24) can be shortened, whereas the second and third sections may be extended. As the main research area is an estuary, I'd expect discussions of carbon cycling on both the freshwater and marine sides, whereas here, only the freshwater side is discussed. I also miss a description of how waters from both sides interact and mix in the estuary, i.e. a section on (seasonality in) C cycling in estuaries.

The authors do not clearly explain in the manuscript why increases in both DIC and TA indicate inputs of $HCO_3^-$, whereas an increase in DIC only must mean an input of $CO_2$. This may not be common knowledge to everyone and should be mentioned in the introduction.

p.6, l.29 - p.7, l.1: I miss some methodological details here. In case surveys were longer than 1 day, was the average discharge for the whole cruise period taken? (this also applies to l.18-20). Plus, I understand that on an annual scale it is valid to assume that discharge at the seawater endmember is the same as riverine discharge, but is this

valid at the time scale of separate surveys (as presented in Table 3) as well? There is another point in the manuscript where these different temporal scales come into play and that is in the context of calculating NEP in section 4.5. If I'm not mistaken, here annual averages for the import and export fluxes are used, whereas it is convincingly shown for at least the import fluxes that there is considerable temporal variability. If the authors did take this into account in their calculations for Fig.9, they should write this more clearly. If they didn't take this into account, I have my doubts about the calculated NEP values.

Section 4.1: Please discuss the reliability and quality of the long-term monitoring data. Such data are often known to display unrealistic trends due to e.g. methodological changes. Also, I do not believe that Fig. 6b displays a real trend as the y-axis variable highly depends on the x-axis variable (as is also shown in Fig. 6d).

Section 4.2, p.9, l-10-13: Don't the authors have enough data available to make a simple linear mixing model at the point where the Schuylkill and Delaware rivers meet near Philadelphia, to actually test and quantify the hypothesis postulated here?

Section 4.3: The authors discuss long-term trends in alkalinity, but as riverine TA export is the product of concentration and discharge, it would be interesting to discuss long-term trends in discharge patterns as well. With the high-resolution data available, the authors can focus not only on long-term trends in discharge, but also on changes in the numbers and intensity of episodic events. Also, the authors disregard the fact that these historical riverine TA data have been previously published and discussed (Kaushal et al., 2013). They should at least refer to this work, and I feel that this manuscript can benefit from the (quantitative) way that work explored possible drivers for the long-term trends. In what has been discussed by the authors, I miss a discussion of the role of increased temperature, which can enhance weathering but has not been shown to the primary driver of weathering in the Baltic Sea catchment (Sun et al., 2017).

p.11, l.12-17: It could be me but this sentence reads like: "Because of X, we assume

X". But, more importantly, the authors do not discuss the validity of their assumption of upscale not only the discharge but also the import fluxes. How valid is it to assume that the remaining 30% of discharge has DIC and TA concentrations equal to the weighted average of the three major rivers?

p.12, l.3: "small riverine systems" No, as these have already been taken into account by upscaling the riverine discharge. I would also suggest to specify groundwater discharge as an additional source here, rather than pooling it into the various external sources.

Section 4.5: I feel that the estimate of NEP can be discussed a bit more in the context of previous work in the estuary. For example, earlier measurements of production and respiration in the estuary also pointed at the latter exceeding the former (Preen and Kirchman, 2004). I am sure there is more relevant work done, perhaps also on the role of salt marshes and groundwater discharge in this system. Also, on p.12, l.27 marshes are mentioned as a possible source of $CO_2$ into the bay, whereas on p.10, l.24-29 it is discussed that the export of DIC from salt marshes is small. So can they really be a substantial $CO_2$ source?

Conclusions: p.13, l. 28-30: The manuscript does not quantify how important seasonal changes in NEP are relative to variations in river discharge and mixing on the same time scale. This ties in with one of my earlier comments on time scales, but would it be possible to show how the relative contribution of NEP versus river discharge changes over the course of the year?

Technical corrections

p.1, l.17: define $HCO_3^-$ before using it.

p.1, l.19: same here for $CO_2$

p.1, l.19-21: this sentence is not very clear. I would at least suggest writing "additional DIC input in the form of $CO_2$" instead of "additional $CO_2$ input", and perhaps do some

more rephrasing.

p.1, l.27: "CO2 flux" should be termed "net DIC production" or, as used later in the manuscript, "net ecosystem production".

p.1, l.27: replace "inclusive of" with "including"

p.1, l.27 - p.2, l.3: It is the small difference between riverine input and export that suggests that most of the DIC produced in situ is lost within the atmosphere, not the fact that in situ production is small to the riverine input. Please rephrase this.

p.2, l.22: add in which form the DIC is transported here (HCO3 or CO2) and whether this depends on silicate versus carbonate weathering.

p.2, l.25: supply of DIC by rivers... add "to estuaries"

p.4, l.8-11: add a short description of the general circulation pattern

p.4, l.13: I miss some basic information here: how many stations were measured each cruise, and what were the coordinates of these stations? The trajectory and stations can easily (and should) be added to Fig. 1.

p.4, l.20: Add a reference to Fig.1 here

p.4, l.22: Add a reference to Fig.2 at the end of the sentence.

p.4, l.28: "preserved at". Also, how long were the samples stored before analysis?

p.5, l.7: What are the accuracy & precision of the pH measurements? What is the potential error with the NBS scale in the more saline waters?

(related to the previous question) p.5, l.18: Here, pH is suddenly mentioned with 3 significant digits, whereas in l.15 and Table 1 only 2 significant digits are given. Please be careful and consistent here.

p.5, l.20: A comma is used here as thousand separator, which is not done in other parts of the manuscript. Please be consistent here.

p.5, l.28-30: I feel this is part of the discussion.

p.6, l.4-6: What do the authors exactly mean with "TA" in l.5? The average concentration of riverine TA? Please clarify.

p.6, l.23: "varied linearly..." add "with salinity"

p.7, l.29: not only respiration from soil OM, but also imbalances between production and respiration along the aquatic continuum can impact DIC:TA ratios.

p.9, l.19: This section should be termed "Historical trends in riverine alkalinity", not "estuarine alkalinity"

p.10, l.9-11: These deviations from conservative mixing for a specific month are really difficult to see in Fig.3.

p.11, l.6 ff.: I'd say that this is a DIC mass balance, not a CO2 mass balance. Please change this throughout the manuscript.

Table 1: Add that pH is at 25 degrees and on the NBS scale. Also, add the in situ temperature such that the reader can get an idea of seasonality in temperature and pH at in situ temperature.

Figure 1: do the arrows point at the exact sampling locations of the rivers? It would be clearer to add symbols indicating the exact locations in the plot. Also, as said before, I miss an indication of the trajectory and/or the exact sampling locations within the estuary in this plot. What is the C&D canal?

Figure 2: add in the caption what the diamond symbols and green lines indicate.

Figure 5: I find it confusing that this plot should be read in the reverse direction as Fig.3 and suggest that the x-axis be reverted.

Figure 6: as discussed above, I suggest removing Fig. 6b (and merge 6d with 6c), as I don't believe it to display a real trend. In the figure caption, change "data measured in

our lab" to "our data". I would also suggest using "our data" in the legends of Figs. 6 and 7, rather than the corresponding author's last name.

References

Kaushal, S. S., Likens, G. E., Utz, R. M., Pace, M. L., Grese, M. and Yepsen, M.: Increased river alkalinization in the eastern U.S., Environ. Sci. Technol., 47(18), 10302–10311, doi:10.1021/es401046s, 2013.

Preen, K. and Kirchman, D.: Microbial respiration and production in the Delaware Estuary, Aquat. Microb. Ecol., 37, 109–119, doi:10.3354/ame037109, 2004.

Sun, X., Mörth, C., Humborg, C. and Gustafsson, B.: Temporal and spatial variations of rock weathering and $CO_2$ consumption in the Baltic Sea catchment, Chem. Geol., doi:10.1016/j.chemgeo.2017.04.028, 2017.

---

## Referee Comment (RC3) · Anonymous Referee #3 · 12 Jul 2017

Joesoef et al. present new data on inorganic carbon seasonal dynamics and land-ocean connectivity from a large North American East Coast estuary that had not previously benefited from significant study of carbonate chemistry (other than the authors' prior publication in this journal a few years ago). Focusing on biogeochemistry, this manuscript is clearly within the scope of BG and will make a nice contribution to the literature after significant revisions to clarify and highlight major concepts and conclusions in the paper.

One newer idea presented in the paper is that the authors draw a distinction between the carbon cycle behavior of larger estuaries compared to the smaller ones that have received more attention in the literature, highlighting the role of different types of habitats (e.g. intertidal wetlands vs. open estuarine water column) in driving the overall

carbon balance of the estuary system. This may benefit from a conceptual diagram if the authors want to argue this is a generalizable phenomenon they are describing – to show the relative influence of different biogeochemical processes and pathways.

While the title is clear and appropriate and the paper is generally well-structured, there are quite a few places where the language is not as clear as it could be. In particular, the abstract could use a fairly substantial rewrite in that the authors' wording is often vague. For example, they say "Our data further suggest that DIC in the Schuylkill River can be substantially different from DIC in the Delaware River, and thus in any river system, tributary contributions must be considered when addressing DIC inputs to the estuary" – the second half of this sentence is so broad as to be essentially meaningless. I'm not sure they ever made a really compelling argument for why this might be important. I am sure there are several reasons why it could be important, but the authors should articulate their reasons for believing this to be important.

The language is fluent and mostly clear, save in a few places where the language becomes vague/imprecise. It may seem redundant to the writer, but there are numerous places where a few more words added would make the difference between vagueness and clarity.

Some examples of these are:

P1, L7: "widely understudied" seems like a bit of a non sequitur/oxymoron.

P1, L26: You might want to say "flux to the coastal ocean" instead of just "flux to the ocean" as in coastal carbon cycle circles, we also discuss export from coastal oceans to the open ocean.

P2, L2: "majority of the DIC produced" – in the estuary, I presume? Clarify.

P2, L10: land-to-ocean continuum?

P2, L 25: "The supply of inorganic carbon by rivers..." – to the coastal ocean?

[Figure]

One part of the interpretation of the data that was never really explained to my satisfaction was why DIC and TA wouldn't both be diluted under higher discharge and thus why the DIC:TA ratio would change with discharge. I suspect that there's a role of temperature in biotic production of CO2 in soils that has a different slope than the temperature dependence of weathering, or something along these lines. The authors' could do a more complete job of illuminating readers on the various factors contributing to the seasonal changes of DIC vs. TA to create a fuller picture and narrative about why they observe a changing DIC:TA ratio through the seasons. The importance of temperature in driving these changes is critical if this work is to have any bearing on predictive studies under future climate change. The discussion of the lithology in the watersheds of each of the study rivers was a bit more detailed than needed, so some of this could be placed into supplemental material, or the text could just be shortened, with the same references. I don't think the detail adds anything to the understanding that one part of the watershed and its tributary contain more carbonate rocks than other parts. Again, why does this matter? (I'm not saying it doesn't, but tell us why you find it important.)

Further comments:

P2, L 23-24: I thought it was just for carbonate minerals that the CO2 is eventually released back to the atmosphere via oceanic carbonate sedimentation, stoichiometrically speaking. Please verify that this statement is correct.

In several places, the authors use the word "impact" when "affect" would be more appropriate. "Impact" is often used to convey negative connotations.

P3, L3: "by weathering and decomposition"

P3, L12: "more large bay systems" would be clearer

P4, L 18: replace "ongoing" with "underway"

P4, L27-28: does this method of filtering samples affect the DIC values? I presume the

references given address this, but if not, it would be good for these authors to address whether filtering samples introduces any artifacts or bias. Filtering DIC samples is not typical (e.g., per the Dickson et al. 2007 SOPs for the CO2 system), but can be done without introducing bias with adequate care (e.g. Bockmon and Dickson 2015? L&O).

P5, L4: Are you sure it's precision that is +/- 2 umol/kg? Vs. some overall uncertainty or average offset from CRMs?

P5, L 5-7: Need to state pH scale is NBS.

P6, L 10: "northernmost"

P6, L21-22: might be good to clarify that this is from the rivers where measurements were taken (vs. the scaled up estimate presented later on).

P6, L27-28: re-cite figure here? Here and just below, it seems like there are a few steps left out of your description of how you did the calculations.

P7, L5: Not enough info given about what this data set is and how the data compare to yours. Put in methods or otherwise describe.

P7, L7: This seasonality doesn't agree with what you described above (summer+fall vs. spring+summer, etc.).

P7, L16: "strong" correlations, not "high" (or "highly correlated")

P7, L20: "dilution of weathering products" (vs. production)

P8, L3: could be faster transport or lower surface area to volume ratio (i.e. deeper)

P9, L1: I am not sure what you mean by "physiographic"

P9, L7: not totally clear what "the historical record" refers to - all USGS data? Just a subset?

P9, L8: closer to 3 decades, at 26 years – maybe "over recent decades" is better?

P9, L9-11: to facilitate reading the paper, it may be best to stick to river names rather than mixing in city names for those readers outside your region

P9, L 16-18: After too much detail on watershed lithology at the start of this section, a bunch of things are summarily mentioned without discussing how these processes might contribute to TA change sufficiently (e.g. would these processes individually increase or decrease TA, and how?).

P9, 26-31: See previous – superficial treatment of these factors (also "can also have huge effects" on following page). Be more specific about the relative roles each of these factors would play if they are important.

P10, L18-20: this is very qualitative. Can you be more quantitative about this?

P11, L12-17: This sentence seems circular to me – how are you defining the difference between input and inflow? If you mean to consider groundwater inputs too, you need to be more concrete and specific with your wording. (Also, there was the roughly 10% from wastewater treatment plants [WWTPs] from up top not mentioned here. Intentional?)

P11, L26: Do you mean water column/internal estuarine CO2 production (per top of next page, I think you do)? Calling it "production" without further clarification of what is being produced gets confusing when primary/community production may also be involved.

P12, L5: near the top, you had a figure for 30 mˆ3/s from WWTPs – this seemed like not a trivial part of the total input

P12, L27: "intertidal" instead of "internal," yes?

P13, L9: "Here" – where are you referring to?

P14, L1: "compared to total DIC input flux" – suggest adding "from rivers"

P26: Seems to ignore interannual variability to list cruises by month w/o noting they

occurred in different years. Do you have enough data on interannual variability to justify that this makes more sense than an alternative? (I don't feel strongly that this shouldn't be done but am curious about the choice to do it this way – would be nice to have some explanation – but doesn't need to be extensive.)

---

## Author Comment (AC1) · 2 Sep 2017

**Author's Response (in Italic)**

*We thank you for the very insightful and constructive reviews, which have greatly enhanced the overall flow and clarity of our manuscript. Following your suggestions, we have made significant modifications and revisions to thoroughly strengthen the presented results and discussion sections. Specifically, we have expanded on the historical USGS data used in this analysis and the methodological uncertainties of this dataset, the comparisons between our data to more comprehensive studies that have investigated long-term alkalinity trends across the U.S. (Kaushal et al., 2013; Stets et al., 2014), and trends in historical river discharge and intensity of episodic events. Further, we now relate our NEP estimates to previous metabolic studies done in the Delaware Estuary (Sharp et al., 1982; Preen and Kirchman, 2004), discuss seasonal variations in NEP, and compare underlying factors that affect DIC mass balance models in large versus small estuarine systems.*

**Referee # 1**

**Major Comments**

Part of the paper is based on the analysis of long-term data-sets from the USGS, going back to the 1940's. The authors compare their own data with the recent USGS (Figure 7), which validates the quality of the recent USGS data. But this does not necessarily mean that the old data are of the same quality, meaning the derived trends over the decades could be methodological. Please add in the discussion, some elements on the methods of TA analysis, data quality check, and any other element that might be useful to show that over the last 70 years the USGS data-set is of uniform quality and that the observed changes are real rather than methodological.

*Response:*

*1) Good point. Frequently, the methodology of scientific methods change over time. To examine this issue, we compared current USGS protocol for alkalinity measurements to previous USGS guidelines and practices to evaluate the impact that methodological changes might have on observed alkalinity trends. Thus, we have added a second paragraph to section '4.3 Historical trends in estuarine alkalinity' as shown below:*

*"While numerous studies across the world indicate a shift towards increasing alkalinity in estuarine waters, the impact of methodological changes on measured values over time cannot be neglected. Conveniently, USGS has published a series of manuals, both past and present, discussing the analytical procedures and methods followed during specialized work in water resources investigations (Woods, 1976; Fishman et al., 1989; Radke et al., 1998). Historically, the USGS measured alkalinity as fixed endpoint titrations on unfiltered samples, and commonly reported values as concentrations of bicarbonate (Clarke, 1924). By 1984, the USGS also began conducting fixed endpoint and incremental titrations on filtered samples (Raymond et al., 2009; Kaushal et al., 2013). Presently, USGS performs several variations of tests that describe the alkalinity of water samples including standard alkalinity, acid neutralizing capacity, and carbonate alkalinity. Samples are measured using either a standard buret, micrometer buret, or*

*by an automated digital titrator (Fishman et al., 1989; Radke et al., 1998). Micrometer burets offer higher accuracy and precision than standard burets while automated titrators are more preferred due to convenience and durability (Radke et al., 1998). Fixed endpoint titrations are generally less accurate than inflection point titrations, especially in low carbonate waters or areas with high organic and noncarbonated contributions to alkalinity (Radtke et al., 2008). Such methodological changes, however, would result in an underestimate of alkalinity if there is any (Kaushal et al., 2013). Thus, our conclusion of an increasing alkalinity trend in the Delaware River water will still hold and can be a conservative estimate. Such alkalinity increase has been observed throughout many river and estuarine systems (Raymond et al., 2003; Raymond et al., 2009; Duarte et al., 2013; Kaushal et al., 2013; Stets et al., 2014)."*

*2) In addition, we described and expanded on the specific USGS alkalinity parameter codes used to clarify water quality data and analytical procedures. We added Table 4 to show the exact parameter codes used during this analysis. The following section was added to the first paragraph in section 4.3.*

*"The extensive and routine collection of water samples conducted by USGS allows us to explore long term trends in alkalinity (from the mid-20[th] to early 21[st] century) in the Delaware and Schuylkill rivers (USGS stations 01463500 and 01474500, respectively). For USGS alkalinity values, we use similar approaches as conducted in Stets et al., 2014. We combine 8 various parameter codes that include alkalinity, acid neutralizing capacity (ANC), or $HCO_3^-$ (Table 4). Alkalinity and ANC follow identical electrometric procedures except that alkalinity samples are filtered while ANC samples are not."*

**Table 4. USGS parameter codes used during analysis**

| Parameter Code | Parameter Description | Total Count | Percentage of Total Count |
| --- | --- | --- | --- |
| 00410 | Acid neutralizing capacity, water, unfiltered, fixed endpoint titration, field | 920 | 28.5 |
| 00419 | Acid neutralizing capacity, water, unfiltered, inflection-point titration, field | 25 | 0.8 |
| 00440 | Bicarbonate, water, unfiltered, fixed endpoint titration, field | 1529 | 47.4 |
| 00450 | Bicarbonate, water, unfiltered, inflection-point titration, field | 25 | 0.8 |
| 00453 | Bicarbonate, water, filtered, inflection-point titration, field | 86 | 2.7 |
| 29801 | Alkalinity, water, filtered, fixed endpoint titration, laboratory | 133 | 4.1 |
| 39086 | Alkalinity, water, filtered, inflection-point titration, field | 283 | 8.8 |
| 90410 | Acid neutralizing capacity, water, unfiltered, fixed endpoint titration, laboratory | 224 | 6.9 |

**Minor Comments**

P2 L 16: to the list of processes that control $CO_2$ in rivers, you could mention inputs from wetlands (Abril et al. 2014).

*Response: Agreed. We have added wetlands to the list of controlling processes as shown below.*

*"The majority of carbon fluxes in inland waters involve inputs from soil-derived carbon, chemical weathering of carbonate and silicate minerals, wetlands, dissolved carbon in sewage waste, and organic carbon produced by phytoplankton in surface waters (Battin et al., 2009; Tranvik et al., 2009; Regnier et al., 2013; Abril et al., 2014)."*

P3 L 10: I assume that this statement is based on some sort of analysis of numbers, so could you please state the range and central value of the area and the residence time of the estuaries from the cited studies.

*Response: Yes, we used the estuarine classification groups as described in Dürr et al., 2011, where surface areas were estimated based on geographical information system analysis. Further, Borges and Abril et al., 2011 listed the criteria for each of the estuarine classification types. We have added the ranges to clarify the differentiation between typical "large" and "small" estuarine systems as shown below.*

*"Further, the majority of past estuarine $CO_2$ studies have focused primarily on small estuarine systems (typically within 1 – 100 km in length and less than 10 m in depth) with rapid freshwater residence times ($10^{-3} – 10^{-1}$ yr) (Chen and Borges, 2009; Cai, 2011; Borges and Abril, 2011, Dürr et al., 2011)."*

P3 L 11: Please define the criteria (threshold value?) and quantity (surface area? discharge? drainage areas? Length?) to distinguish "large" and "small" estuaries.

*Response: Please see above.*

P7 L 12-13: The correlation between TA fluxes and discharge is due to auto-correlation. If you plot AxB versus B, you'll always generate a good correlation (Berges 1997), especially if B changes over several orders of magnitude (unlike A).

*Response: We are aware of this issue and its potential concern. As TA flux is defined as concentration multiplied by discharge, one would expect a solid correlation between the two variables. We offer the following explanation to clarify our approach. Here, in Fig. 4 we plot TA against river discharge to determine the correlation between the two variables (strong negative correlation). We used this relationship paired with high frequency USGS discharge records to estimate high resolution TA at the river end-members. Then, we calculated the annual TA flux. We did not directly use F= TA*Q to plot against Q to determine the annual TA flux. We believe the difference between our approach and what the reviewer has mentioned as auto-correlation is that our statistically significant correlations in Fig. 4 is between TA and discharge and not*

*between flux and discharge (subsequently the regression is not directly driven by discharge, though geochemically they are related).*

P 10 L 25: The finding that intertidal marshes have little influence on the $CO_2$ dynamics of the Delaware is quite interesting and would contradict the main conclusion and among the opening statements of the Cai (2011) paper: "It is demonstrated here that $CO_2$ release in estuaries is largely supported by microbial decomposition of highly productive intertidal marsh biomass".

*Response: Agreed. In our revision, we further expand on the interesting contrasts with the small southern estuaries emphasized in Cai (2011). The impact of intertidal marshes on estuarine $CO_2$ dynamics can be significant, particularly in small estuarine systems. In this study, we did not sample the sub-estuaries within nor areas near the perimeters of the bay, but instead were limited to sampling within the main channel of the estuary. We note while the Delaware River is only a medium size river, the Delaware Bay is one of the largest bays in the U.S. eastern coast and its hydrodynamics is largely controlled by the exchange with the ocean (residence time of 1-3 months). Previous studies that conducted cross bay transects, sampling at various depths, over diel cycles, and along tributaries, found that except near the shoreline where suspended sediment and chlorophyll concentrations were high, general cross-bay gradients were erratic and comparatively small (Culberson et al., 1987; Lebo et al., 1990; Sharp et al., 2009). While significantly more research and data are needed, we suggest that due to the much broader geographical size of the Delaware Bay, that except near shallow waters the flushing of intertidal marshes has a minor impact on overall surface water $pCO_2$ and $CO_2$ flux dynamics in the system (as opposed to in small estuaries where marshes may have significant impacts on estuarine $CO_2$ degassing fluxes).*

P 13 L 6: This discussion seems to contradict the Introduction (P3 L10) that previously studied estuaries have a "short residence time"

*Response: We see why this might illustrate contrasting ideas and perhaps we need to restructure/reemphasize key points to clarify the main objective of this paragraph. In the Introduction, we stated that most of the previously studied estuaries have a "short residence time" but the Scheldt is an exception. Here, we want to contrast the differences in ecosystem metabolism between estuaries with short versus long residence times. The Scheldt Estuary has a long, freshwater residence time similar to that of the Delaware Bay. We would like to use prior metabolic findings in the Scheldt Estuary to serve as a model for net ecosystem metabolism in the Delaware estuary and potentially other large estuarine systems with long residence times. We revised our text to make this point clear.*

*"Unlike in most previously studied estuaries, freshwater residence times in the Scheldt Estuary and Delaware Bay are generally long ranging from about one to a few months (Gay and O'Donnell, 2009; Borges and Abril, 2011) ... Thus, we suspect that in other estuarine systems with long freshwater residence times (i.e. the Delaware Estuary), much of the DIC produced by NEP is most likely removed to the atmosphere rather than exported to the sea."*

Figure 1: axis legends have a different font from all of the other figures, it is advisable to have a uniform font in all figures.

*Response: Good point. We have revised it so that all fonts are uniform.*

Figure 4: Two decimals for $R^2$ are sufficient. In some figures, numbers in axis legend have thousands separated by comma, but not in others. It is advisable to make this uniform. In some figures, the axis name is "alkalinity", in others it is "TA". It is advisable to make this uniform.

*Response: We have reduced to two decimal points for $R^2$ values. We also fixed the number issue in the axis legends and used 'alkalinity' as the uniform axis name in all figures.*

Figures 6 and 7: It is odd that one of the data-sets is named after one of the authors ("Cai"), I suggest that the data set is named "this study", something neutral and a bit more modest.

*Response: Agreed. We have changed it to "This Study".*

**Referee # 2**

**Major Comments**

I found the introduction particularly unbalanced. Specifically, I think that the first paragraph of the introduction (p. 2, l. 7-24) can be shortened, whereas the second and third sections may be extended. As the main research area is an estuary, I'd expect discussions of carbon cycling on both the freshwater and marine sides, whereas here, only the freshwater side is discussed. I also miss a description of how waters from both sides interact and mix in the estuary, i.e. a section on (seasonality in) C cycling in estuaries.

*Response: Good point. We have shorten the first paragraph of the introduction and added more discussion on carbon cycling in estuaries, specifically on the marine sides. Please see below.*

*"As carbon is transported horizontally along the land and ocean continuum, various environmental processes impact the total carbon fluxes between reservoirs. Recent studies suggest that on average 10% of $CO_2$ emitted in estuaries is sustained by freshwater inputs while 90% of the $CO_2$ released is from local net heterotrophy, with the majority of organic carbon inputs stemming from adjacent salt marsh and mangrove ecosystems (Regnier et al., 2013). These systems are supported by inputs from various autochtonous and allochtonous organic carbon sources, $CO_2$ enriched pore waters during ebbing, and high concentrations of dissolved inorganic carbon from inter-tidal and sub-tidal benthic communities (Cai et al., 2003; Neubauer and Anderson, 2003; Wang and Cai, 2004; Ferrón et al., 2007; Chen and Borges, 2009). Terrestrial OC that is transported by large and fast-transit river systems generally bypasses decomposition in estuaries and contributes to respiration along coastal ocean margins (Cai, 2011). Consequently, rapid increases in atmospheric $CO_2$ concentrations reduce the amount of $CO_2$ released along ocean margin systems, especially in low latitude zones where a majority of the terrestrial OC is delivered (Cai, 2011)."*

The authors do not clearly explain in the manuscript why increases in both DIC and TA indicate inputs of $HCO_3^-$, whereas an increase in DIC only must mean an input of $CO_2$. This may not be common knowledge to everyone and should be mentioned in the introduction.

*Response: We agree that this is unclear and may not be common knowledge. In turn, we should define and expand on these terms in the introduction. We have added the following text to paragraph two of the introduction.*

*"Total alkalinity (TA) is defined as TAlk = $[HCO_3^-]$ + $2[CO_3^{2-}]$ plus all other weak bases that can accept $H^+$ when titrated to the carbonic acid endpoint. Comparably, dissolved inorganic carbon (DIC) is expressed as the sum of all inorganic carbon species ($[CO_2]$, $[HCO_3^-]$, $[CO_3^{2-}]$). In terrestrial aquatic systems, there are three sources of dissolved inorganic carbon. The most important sources are the carbonate and silicate weathering processes as described below:*

*$CaCO3 + CO2 \rightarrow 2HCO3- + Ca2+$*
*$CaSiO3 + 2CO2 + 3H2O \rightarrow 2HCO3- + Ca2+ + H4SiO4$*

*In both cases, the amounts of DIC and TA production are equal. Here, $CO_2$ may come from soil organic matter respiration but ultimately it is linked to the atmospheric $CO_2$. Respiration of soil and aquatic organic carbon is another source of $CO_2$, but it does not contribute to TA. Since alkalinity of natural waters is mainly comprised of $[HCO_3^-]$ and $[CO_3^{2-}]$ ions and all other species are generally insignificant, DIC to TA ratios can provide broad insight into the sources of carbon, aquatic pH dynamics and regional carbonate buffering capacity."*

p.6, l.29 - p.7, l.1: I miss some methodological details here. In case surveys were longer than 1 day, was the average discharge for the whole cruise period taken? (this also applies to l.18-20). Plus, I understand that on an annual scale it is valid to assume that discharge at the seawater endmember is the same as riverine discharge, but is this valid at the time scale of separate surveys (as presented in Table 3) as well? There is another point in the manuscript where these different temporal scales come into play and that is in the context of calculating NEP in section 4.5. If I'm not mistaken, here annual averages for the import and export fluxes are used, whereas it is convincingly shown for at least the import fluxes that there is considerable temporal variability. If the authors did take this into account in their calculations for Fig.9, they should write this more clearly. If they didn't take this into account, I have my doubts about the calculated NEP values.

*Response: Yes, we used the average discharge for the whole cruise period to estimate input fluxes during this time. We have added this detail to our flux calculation descriptions. You bring up an interesting point in our export flux calculations about how it is valid to assume that seawater endmember discharge is equivalent to riverine discharge on an annual time scale, but it may not be for the time scales of separate surveys. We did take this into account and used average discharge for the entire cruise period plus discharges recorded 10 days prior to the survey. We agree that this approach must have substantial uncertainty but feel it is probably a good first order approximation given largely linear distribution of DIC at high salinity.*

Section 4.1: Please discuss the reliability and quality of the long-term monitoring data. Such data are often known to display unrealistic trends due to e.g. methodological changes. Also, I do not believe that Fig. 6b displays a real trend as the y-axis variable highly depends on the x-axis variable (as is also shown in Fig. 6d).

*Response: Agreed. Often, the methodology of scientific methods change over time, especially with advancements in technology. We have added some discussion investigating these changes to section '4.3 Historical trends in river alkalinity'. Please see our response to Referee #1.*

Section 4.2, p.9, l-10-13: Don't the authors have enough data available to make a simple linear mixing model at the point where the Schuylkill and Delaware rivers meet near Philadelphia, to actually test and quantify the hypothesis postulated here?

*Response: Agreed. In fact, we did use a simple three end-member mixing model to estimate the composite river DIC and TA concentrations at the confluence of the Delaware and Schuylkill river ($C_m^*$), and then using this value determined the composite concentrations at the confluence of the Delaware and Christina rivers ($C_{m2}^*$) as well. We multiplied ($C_{m2}^*$) by total river discharge*

*($Q_T$) to compute riverine input fluxes. We agree that these steps were missing in the manuscript and should be added to avoid any confusion as shown below:*

*"Water mixing from multiple tributaries can complicate two end-member mixing models (Officer, 1979; Cai et al., 2004). In this case, from Trenton, NJ to the mouth of the Delaware Bay, two external sources of water, the Schuylkill and Christina River, discharge into the Delaware River. Thus, additional input from these tributaries contribute to effective end-member concentrations. Past studies have shown that a composite river end-member can be estimated for a set of tributaries given their respective discharge rates (Cai et al., 2004; Guo et al., 2008). In turn, a simple three end-member mixing model can be generated where $C_1$, $C_2$, and $C_s$ represent the end-member concentrations at the Delaware River, the Schuylkill River, and the ocean side, respectively. Assuming that only mixing occurs between the two river end-members, we can estimate a new effective concentration ($C_m^*$) as follows:*

$$C_m^* = \frac{C_1 \times Q_1 + C_2 \times Q_2}{Q_1 + Q_2}, \tag{1}$$

*where $Q_1$ and $Q_2$ represent discharge rates for rivers 1 and 2, respectively. The linear mixing line for $C_{1\text{-}s}$ is estimated as follows:*

$$C_{1\text{-}s} = C_1 + \frac{C_s - C_1}{S_s} S, \tag{2}$$

*Linear mixing equations for $C_{2\text{-}s}$ are similar. Likewise, $C_m$ is a linear combination of $C_{1\text{-}s}$ and $C_{2\text{-}s}$ and is estimated as follows:*

$$C_m = C_m^* + \frac{C_s - C_m^*}{S_s} S, \tag{3}$$

*Through this simple three end-member mixing model, we estimate a composite river end-member at the confluence of the Delaware and Schuylkill River (approximately 150km from the mouth of the Delaware Bay). The chemical fluxes for each tributary can be calculated as follows:*

$$F_i = C_i \times Q_i, \tag{4}$$

*Thus, the total flux at the confluence of the Schuylkill and Delaware River is estimated as:*

$$F^T = C_m^* \times Q^T = F_1 + F_2, \tag{5}$$

*With $C_m^*$ as a new upstream end-member value, we can further estimate the composite river end-member ($C_{m2}^*$) at the confluence of the Delaware and Christina River (approximately 110km from the mouth of the bay) using end-member concentrations and discharge rates for the Christina River and the previous equations above. To compute the total river input flux, we add estimated flux from the Christina River ($C_3 \times Q_3$) to the results obtained from Eq. 5 as shown below:*

$$F^T = C_m^* \times Q^T + (C_3 \times Q_3) = F_1 + F_2 + F_3, \tag{6}$$

*or $F^T = C^*_{m2} \, x \, Q^T$ where $C^*_{m2}$ is calculated from Eq. 1 with $C^*_m$ and $C_3$ being the two river end-members, $Q_1$ being the sum of the Delaware and Schuylkill River discharge, and $Q_2$ being the discharge rate for the Christina River."*

Section 4.3: The authors discuss long-term trends in alkalinity, but as riverine TA export is the product of concentration and discharge, it would be interesting to discuss long term trends in discharge patterns as well. With the high-resolution data available, the authors can focus not only on long-term trends in discharge, but also on changes in the numbers and intensity of episodic events. Also, the authors disregard the fact that these historical riverine TA data have been previously published and discussed (Kaushal et al., 2013). They should at least refer to this work, and I feel that this manuscript can benefit from the (quantitative) way that work explored possible drivers for the long-term trends. In what has been discussed by the authors, I miss a discussion of the role of increased temperature, which can enhance weathering but has not been shown to the primary driver of weathering in the Baltic Sea catchment (Sun et al., 2017).

*Response:*

*1) Agreed. With such high-resolution data, it would be informative to examine long-term trends in discharge in addition to long-term trends in river alkalinity. We have added additional figures plotting daily mean discharge recorded in the Delaware and Schuylkill rivers from 1940 to 2015. We further highlight the intensity of episodic discharge events at these locations (defined by the average daily discharge plus 10 standard deviations). We also added the following section '4.4 Long-term trends in river discharge' to the discussion portion of the paper as shown below.*

*"To investigate long-term trends in discharge for the Delaware and Schuylkill rivers, we plot daily discharge from 1940 – 2015 at Trenton, NJ and Philadelphia, PA. Further, we follow similar methods as discussed in Voynova and Sharp (2012) to examine the intensity of episodic discharge events (defined by the average daily discharge plus 10 standard deviations) with time (Fig. 8). Unlike historical trends in river alkalinity, there has been minimal to no increase in mean discharge over time in the Delaware and Schuylkill rivers suggesting that increased alkalinity flux is due to increased alkalinity concentrations and weathering rates. While there was no long-term increase in mean river discharge, the frequency of episodic events with time has significantly increased. Over the past 70 years, 29 extreme discharges have been recorded in the Schuylkill River (from 1 Jan 1940 to 31 Dec 2015) with 48% of these occurring in the past two decades. Similarly, recent study by Voynova and Sharp (2012) showed that in the past century 54 extreme discharges have been recorded in the Delaware River (from 1 Oct 1912 to 30 Sept 2011). Of the 54 extreme discharges, 46% of these occurred during the past decade. Bauer et al., 2013 suggest that episodic discharge events (large flooding/heavy rains) can carry a disproportionately large part of the annul flux of organic carbon from a certain drainage basin. Our work suggests that this mechanism may also apply to riverine TA flux. Thus, with recent evidence indicating a shift towards more frequent episodic weather events, it is important to consider how such anomalies impact biogeochemical patterns among coastal systems (i.e. prolonged summer stratification, freshwater residence times, riverine bicarbonate concentrations, estuarine $CO_2$ fluxes) (Allan and Soden, 2008; Yoana and Sharp, 2012)."*

*2) We agree that we should refer to previous work done by Kaushal et al., 2013 as this would greatly enhance and support our discussion on increasing trends in riverine alkalinity. In addition, we also should refer to previous work done by Stets et al., 2014 as they conducted a similar study investigating long-term alkalinity trends in river systems throughout the U. S. We have revised and expanded on this section to incorporate the following:*

*"A more comprehensive study conducted by Kaushal et al., 2013 examined long-term trends in river alkalinity at 97 different stream and river locations throughout the eastern U.S. They observed increasing alkalinity trends at 62 of the 97 river locations (64%). Moreover, of the remaining sites, none showed any statistically decreasing alkalinity trends. Various contributing factors can influence long-term trends in river alkalinity such as carbonate lithology, acid deposition, and topography in watersheds. Kaushal et al., 2013 suggests that increased acid deposition elevates riverine alkalinity by promoting weathering processes, particularly in watersheds with high carbonate lithology. Further, watershed elevation may be a good predictor for alkalization rates. Acid deposition may be greater at higher elevations, and such areas tend to have thinner soils and a weaker buffering capacity, increasing susceptibility to the effects of acid deposition. Recent studies show that human induced land-use changes such as deforestation, agricultural practices (Oh and Raymond, 2006), and mining activities (Brake et al., 2001; Raymond and Oh, 2009) have direct impacts on the buffering capacity of streams and rivers. Through chemical weathering processes, enhanced precipitation and local runoff can also have huge effects on increased alkalinity in coastal ecosystems (Raymond et al., 2008). For example, it was suggested that over the past century, total alkalinity export from the Mississippi River to the Gulf of Mexico has risen by nearly 50% due to widespread cropland expansion and increased precipitation in the watershed (Raymond and Cole, 2003; Raymond et al., 2008). Comparably, Stets et al., 2014 explored historical time series of alkalinity values in 23 different riverine systems throughout the U.S. They found increasing alkalinity trends at 14 of these locations with the majority occurring in the Northeastern, Midwestern, and Great Plains areas of the U.S. While most sites observed increasing alkalinity values with time, decreasing trends were found in the Santa Ana, Upper Colorado, and Brazos rivers. Factors contributing to decreasing trends at these locations include dilution by water from external sources outside the basin and retention of weathering products in storage reservoirs."*

p.11, l.12-17: It could be me but this sentence reads like: "Because of X, we assume X". But, more importantly, the authors do not discuss the validity of their assumption of upscale not only the discharge but also the import fluxes. How valid is it to assume that the remaining 30% of discharge has DIC and TA concentrations equal to the weighted average of the three major rivers?

*Response: Good point. We have acknowledged this problem. We also changed "estimate" to "assume". Yes, you are correct in that we should discuss the validity of our assumptions and highlight the possible errors in to our calculations. However we must point out as such upscaling applies to both input (river flux) and export (estuarine flux to the offshore), it doesn't affect our conclusion on the DIC source and sink balance. We have added the following explanation to the DIC mass balance discussion section.*

*"Since approximately 70% of the freshwater inflow to the estuary comes from the Delaware, Schuylkill, and Christina rivers, and the remaining percentage comes from small rivers and nonpoint source runoff, we estimate that the Delaware, Schuylkill, and Christina rivers provide the estuary with about 70% (annual mean discharge of these rivers together was 387 $m^3$ $s^{-1}$ from 2013-2015) of its total freshwater input. Thus, by upward scaling, we obtain an annual mean discharge of 553 $m^3$ $s^{-1}$ and a final DIC input flux of 15.7 $\pm$ 8.2 $\times$ $10^9$ mol C $yr^{-1}$ and export flux of 16.5 $\pm$ 10.6 $\times$ $10^9$ mol C $yr^{-1}$. It is important to note that these final flux values are a rough estimate.*

*"We acknowledge that average riverine DIC and TA concentrations from remaining small rivers and nonpoint source runoff are not necessarily equivalent to the weighted DIC and TA averages for the Delaware, Schuylkill, and Christina rivers. As such uncertainties are most often neglected, it is imperative to consider their impact on final flux values. However, since extensive research and data is needed, here we assume that the mineralogy and drainage basins of the remaining small rivers yield similar carbonate concentrations as Delaware's three major river systems."*

p.12, l.3: "small riverine systems" No, as these have already been taken into account by upscaling the riverine discharge. I would also suggest to specify groundwater discharge as an additional source here, rather than pooling it into the various external sources.

*Responses: Good catch. After upscaling the riverine discharge, we eliminated additional input from small riverine systems (i.e. creeks). We also removed this variable from our DIC mass balance equation and specified benthic recycle and ground water discharge as an additional source. The following section now reads as follows:*

*"Thus, a DIC mass balance for the estuary is formed as follows:*

*River input flux (15.7 $\times$ $10^9$ mol C $yr^{-1}$)*
*+ Internal estuarine production (?)*
*+ Inputs from surrounding salt marshes (?)*
*+ Inputs form benthic recycling (?)*
*= Estuarine output flux (16.5 $\times$ $10^9$ mol C $yr^{-1}$)*
*+ Atmospheric flux (4.3 $\times$ $10^9$ mol C $yr^{-1}$)*

*The total sum of the unknown internal DIC production terms is therefore estimated as 5.1 $\times$ $10^9$ mol C $yr^{-1}$. This total internal DIC production includes respiration in the water column and benthos, $CO_2$ addition from intertidal marsh waters, wastewater effluents, ground water discharge, and other various external sources. If we pool water column and benthic respiration into one term and ignore additional input from wastewater effluents and ground water discharge, DIC fluxes can be viewed as a measure of net ecosystem production (NEP)."*

Section 4.5: I feel that the estimate of NEP can be discussed a bit more in the context of previous work in the estuary. For example, earlier measurements of production and respiration in the estuary also pointed at the latter exceeding the former (Preen and Kirchman, 2004). I am sure there is more relevant work done, perhaps also on the role of salt marshes and groundwater

discharge in this system. Also, on p.12, l.27 marshes are mentioned as a possible source of $CO_2$ into the bay, whereas on p.10, l.24-29 it is discussed that the export of DIC from salt marshes is small. So can they really be a substantial $CO_2$ source?

Conclusions: p.13, l. 28-30: The manuscript does not quantify how important seasonal changes in NEP are relative to variations in river discharge and mixing on the same time scale. This ties in with one of my earlier comments on time scales, but would it be possible to show how the relative contribution of NEP versus river discharge changes over the course of the year?

*Responses:*

*1) Agreed. Additional comparisons to previous work done in the estuary would significantly strengthen the paper's discussion. We have expanded the section to discuss more about previous studies that have investigated respiration, production, and net ecosystem production within the estuary as shown below.*

*"While this study estimates overall NEP of the Delaware Estuary, other studies have explored NEP across the estuarine gradient (Sharp et al., 1982; Lipschultz et al., 1986; Hoch and Kirchman, 1993; Preen and Kirchman, 2004). Significant depletion of dissolved oxygen and supersaturation of $pCO_2$ levels in freshwaters (salinity < 10), suggests that the upper estuary is heterotrophic while the lower estuary is autotrophic (Sharp et al., 1982). More recent studies have found that respiration often exceeds primary production in the upper Delaware River (Hoch and Kirchman, 1993; Preen and Kirchman, 2004). Comparably, Culberson (1988) used inorganic carbon and dissolved oxygen measurements to estimate apparent carbon production and oxygen utilization throughout the Delaware Estuary. Similar to our spring NEP results, Culberson (1988) found that during the months of March to May from 1978 to 1985, most of the estuary (6 < S < 30) suffered a net inorganic carbon loss. Presumably, this loss occurred during the spring phytoplankton bloom, a period of intense inorganic carbon uptake by phytoplankton. While respiration rates often outweigh primary production in the upper tidal river, generally net community production increases down the estuary, transitioning to a near balanced to autotrophic system in the mid- to lower bay regions (Hoch and Kirchman, 1993; Preen and Kirchman, 2004)."*

*2) In regards to additional $CO_2$ input from surrounding marsh systems, we agree that intertidal marshes can have drastic impacts to estuarine $CO_2$ dynamics, particularly in small estuarine systems. However, previous work found that in general cross-bay gradients were erratic and comparatively small (Culberson et al., 1987; Lebo et al., 1990; Sharp et al., 2009), consistent with our main channel bay study that found marsh impact to be small. In our case, significantly more research and data are needed especially near the perimeters of the estuary to accurately ascertain the impact from marsh systems. Thus, we caution the audience in jumping to conclusions as it is unclear whether the organic matter respiration occurs in the main channel of the estuary or from nearby internal marshes with the resulting $CO_2$ flushed into the bay.*

*3) Yes, it is important to examine the relationship between NEP and variations in seasonal discharge. We have added additional discussion comparing the two variables to section 4.5 as*

*shown below. We also added an additional figure comparing seasonal discharge, NEP, and air-water CO₂ fluxes.*

*"Riverine input and estuarine export fluxes show considerable temporal variability and are largely governed by seasonal discharge patterns (Table 2 and 3). The highest fluxes occurred during spring when discharge was high while the lowest values occurred in the fall and winter when discharge was low. However, seasonal changes in NEP did not reflect variations in river discharge. Discharge values decreased throughout the year while NEP rates fluctuated across seasons. On the other hand, NEP trends largely mirrored seasonal variations in air-water CO₂ fluxes. When the estuary acted as a source of CO₂, negative NEP was observed. In comparison, when the system served as a CO₂ sink, NEP was positive. From the annual mass balance model, the small difference between riverine input and export flux suggests that the majority of DIC produced within the estuary is exchanged with the atmosphere rather than exported to the ocean. It is important to note that such conclusions were estimated based on surveys conducted during different months from different years. More research and data is needed to accurately ascertain seasonal variations in estuarine fluxes and NEP."*

**Minor Comments**

p.1, l.17: define $HCO_3^-$ before using it.
p.1, l.19: same here for $CO_2$
p.1, l.19-21: this sentence is not very clear. I would at least suggest writing "additional DIC input in the form of $CO_2$" instead of "additional $CO_2$ input", and perhaps do some more rephrasing.

*Response: We agree. We have defined $HCO_3^-$ and $CO_2$ and rephrased the sentence as follows, "The ratio of DIC to TA, an understudied but important property, is high (1.11) during high discharge and low (0.94) during low discharge, reflecting additional DIC input in the form of carbon dioxide ($CO_2$), most likely from organic matter decomposition rather than from other bicarbonate ($HCO_3^-$) inputs due to drainage basin weathering processes."*

p.1, l.27: "$CO_2$ flux" should be termed "net DIC production" or, as used later in the manuscript, "net ecosystem production".
p.1, l.27: replace "inclusive of" with "including".
p.1, l.27 - p.2, l.3: It is the small difference between riverine input and export that suggests that most of the DIC produced in situ is lost within the atmosphere, not the fact that in situ production is small to the riverine input. Please rephrase this.

*Response: Good points. We have rephrased the sentences as follows, "Annual DIC input flux to the estuary and export flux to the ocean are estimated to be $15.7 \pm 8.2 \times 10^9$ mol C yr$^{-1}$ and $16.5 \pm 10.6 \times 10^9$ mol C yr$^{-1}$, respectively, while net DIC production within the estuary including inputs from intertidal marshes is estimated to be $5.1 \times 10^9$ mol C yr$^{-1}$. The small difference between riverine input and export flux suggest that, in the case of the Delaware Estuary and perhaps other large coastal systems with long freshwater residence times, the majority of the DIC produced by biological processes is exchanged with the atmosphere rather than exported to the sea."*

p.2, l.22: add in which form of DIC is transported here ($HCO_3^-$ or $CO_2$) and whether this depends on silicate versus carbonate weathering.

*Response: We agree. We should be more specific here. We have added the weathering reactions in the introduction and modified the sentence to, "The weathering of carbonate and silicate minerals consumes atmospheric $CO_2$ and transports $HCO_3^-$ ions and subsequent cation and anion products into oceanic systems. Eventually, $CO_2$ is released back into the atmosphere via oceanic carbonate sedimentation and volcanic activity (Lerman et al., 2004; Regnier et al., 2013)."*

p.2, l.25: supply of DIC by rivers…add "to estuaries".

*Response: Added. Now read as, "Typically, the supply of inorganic carbon by rivers to estuaries is governed by river discharge, weathering intensity, and the geology of the drainage basin (White and Blum, 1995; White, 2003; Guo et al., 2008)."*

p.4, l.13: I miss some basic information here: how many stations were measured each cruise, and what were the coordinates of these stations? The trajectory and stations can easily (and should) be added to Fig. 1.

*Response: Unfortunately, most of the research cruises were conducted on ships of opportunity (i.e. funding was supported by other lab groups). Thus, the number of sampling stations and cruise path varied throughout surveys. In turn, for our study we collected surface water CTD and underway samples across the salinity gradient. Since stations were different for each cruise, it is difficult for us to label their locations in Fig. 1 and we have clarified this in the text as shown below:*

*"DIC, TA, and pH were measured along the salinity gradient of the Delaware Estuary on eight cruises: 8-10 June 2013, 17-22 November 2013, 23-24 March 2014, 2-3 July 2014, 27 August to 1 September 2014, 30 October to 2 November 2014, 5 December 2014, and 6 April 2015. However, because stations were different for each cruise, we do not label them in Fig. 1."*

p.4, l.20: Add a reference to Fig.1 here.
p.4, l.22: Add a reference to Fig.2 at the end of the sentence.

*Response: References have been added.*

p.4, l.28: "preserved at". Also, how long were the samples stored before analysis?

*Response: Corrected. Now reads as, "DIC and TA samples were filtered through a cellulose acetate filter (0.45 μm) into 250 ml borosilicate bottles, fixed with 100 μl of saturated mercury chloride solution, preserved at 4°C, and analyzed within two weeks of sample collection (Cai and Wang, 1998; Jiang et al., 2008)."*

p.5, l.7: What are the accuracy & precision of the pH measurements? What is the potential error with the NBS scale in the more saline waters?

(related to the previous question) p.5, l.18: Here, pH is suddenly mentioned with 3 significant digits, whereas in l.15 and Table 1 only 2 significant digits are given. Please be careful and consistent here.

*Response: We agree. We should be consistent when reporting measurement values. In this study, pH values were measured to within 0.005 units however the expected accuracy is probably not better than 0.01 units. We have changed all pH values to two significant figures. In addition, our methods now read as, "For pH measurements, water samples were collected in glass bottles with a narrow mouth and left in a thermal bath (at 25°) for about 30-60 minutes. pH was then determined onboard using an Orion 3-Star Plus pH Benchtop Meter with a Ross pH electrode (Thermo Fisher Scientific Inc. Beverly, MA, USA) and calibrated using three National Bureau Standard (NBS) pH buffers of 4.01, 7.00, and 10.01. Note that the narrow mouth of the glass bottle is only slightly larger than the outer diameter of the pH electrode thus preventing $CO_2$ degassing during the analysis. While the analytical precision is ±0.005 units, the expected accuracy is probably not better than ±0.01 pH units."*

p.5, l.20: A comma is used here as thousand separator, which is not done in other parts of the manuscript. Please be consistent here.

*Response: Yes. Consistency is key and we have removed this accordingly.*

p.5, l.28-30: I feel this is part of the discussion.

*Response: We agree and have moved these sentences to section '4.1 Influence of river discharge and weathering intensity'.*

p.6, l.4-6: What do the authors exactly mean with "TA" in l.5? The average concentration of riverine TA? Please clarify.

*Good point. We have clarified on this description. Now reads as, "Despite mixing from multiple end-members, such differences in discharge indicate that average riverine TA is predominantly governed by carbonate concentrations in the Delaware River."*

p.6, l.23: "varied linearly…" add "with salinity".

*Added.*

p.7, l.29: not only respiration from soil OM, but also imbalances between production and respiration along the aquatic continuum can impact DIC:TA ratios.

*Agreed. The sentence now read as follows, "On the other hand, $CO_2$ production from soil organic matter respiration and imbalances between production and respiration along the aquatic continuum can increase DIC to TA ratios (Mayorga et al., 2005)."*

p.9, l.19: This section should be termed "Historical trends in riverine alkalinity", not

"estuarine alkalinity".

*Changed.*

p.10, l.9-11: These deviations from conservative mixing for a specific month are really difficult to see in Fig.3.

*Yes, perhaps there is a better way to display this variation. Plotting separate months may be better to see individual trends, however we would lose the group comparison gained when plotting all months together.*

p.11, l.6 ff.: I'd say that this is a DIC mass balance, not a $CO_2$ mass balance. Please change this throughout the manuscript.

*Good point. We have changed this throughout the manuscript.*

Table 1: Add that pH is at 25 degrees and on the NBS scale.

*We have added that pH was measured at 25 degrees on the NBS scale.*

Figure 1: do the arrows point at the exact sampling locations of the rivers? It would be clearer to add symbols indicating the exact locations in the plot. Also, as said before, I miss an indication of the trajectory and/or the exact sampling locations within the estuary in this plot. What is the C&D canal?

*Good points. We now describe what the black arrows mean (river names) in Fig.1. As mentioned earlier, most of the research cruises were conducted on ships of opportunity. Thus, the number of sampling stations and cruise path varied throughout surveys and would be difficult to plot in the figure. In turn, for our study we collected surface water CTD and underway samples across the salinity gradient. We have removed C&D canal as it is not necessary.*

Figure 2: add in the caption what the diamond symbols and green lines indicate.

*Yes, we now describe that the red diamonds indicate exact sampling dates and the green lines are when river waters were frozen.*

Figure 5: I find it confusing that this plot should be read in the reverse direction as Fig. 3 and suggest that the x-axis be reverted.

*Good idea. We have reverted the axis direction so that it is more comparable to our previous plots.*

Figure 6: as discussed above, I suggest removing Fig. 6b (and merge 6d with 6c), as I don't believe it to display a real trend. In the figure caption, change "data measured in our lab" to "our data". I would also suggest using "our data" in the legends of Figs. 6 and 7, rather than the corresponding author's last name.

*Interesting point. As TA flux is defined as concentration multiplied by discharge, one would expect a solid correlation between the two variables. We have removed Fig. 6b and merged 6c and 6d. In the figure caption, we changed 'data measured in our lab' to 'our data'. Yes, for the legend description we have changed the name to 'This Study' instead of 'Cai' in Fig. 6 and 7.*

**Referee # 3**

**Major Comments**

One newer idea presented in the paper is that the authors draw a distinction between the carbon cycle behavior of larger estuaries compared to the smaller ones that have received more attention in the literature, highlighting the role of different types of habitats (e.g. intertidal wetlands vs. open estuarine water column) in driving the overall carbon balance of the estuary system. This may benefit from a conceptual diagram if the authors want to argue this is a generalizable phenomenon they are describing – to show the relative influence of different biogeochemical processes and pathways.

*Response: Good point. We have expanded on the comparison and discussion of NEP and DIC mass balances between various estuarine systems. We now compare our results to previous studies that investigated respiration and production rates throughout the Delaware Estuary. Further, we have expanded on the impact of discharge on seasonal variations in NEP and discuss how different physical features can affect overall DIC production within estuarine systems. The following revised/modified parts of discussion section 4.6 'DIC mass balance' are shown below:*

*"While this study estimates overall NEP of the Delaware Estuary, other studies have explored NEP across the estuarine gradient (Sharp et al., 1982; Lipschultz et al., 1986; Hoch and Kirchman, 1993; Preen and Kirchman, 2004). Significant depletion of dissolved oxygen and supersaturation of $pCO_2$ levels in freshwaters (salinity < 10), suggests that the upper estuary is heterotrophic while the lower estuary is autotrophic (Sharp et al., 1982). More recent studies have found that respiration often exceeds primary production in the upper Delaware River (Hoch and Kirchman, 1993; Preen and Kirchman, 2004). Comparably, Culberson (1988) used inorganic carbon and dissolved oxygen measurements to estimate apparent carbon production and oxygen utilization throughout the Delaware Estuary. Similar to our spring NEP results, Culberson (1988) found that during the months of March to May from 1978 to 1985, most of the estuary (6 < S < 30) suffered a net inorganic carbon loss. Presumably, this loss occurred during the spring phytoplankton bloom, a period of intense inorganic carbon uptake by phytoplankton. While respiration rates often outweigh primary production in the upper tidal river, generally net community production increases down the estuary, transitioning to a near balanced to autotrophic system in the mid- to lower bay regions (Hoch and Kirchman, 1993; Preen and Kirchman, 2004)."*

*"Riverine input and estuarine export fluxes show considerable temporal variability and are largely governed by seasonal discharge patterns (Table 2 and 3). The highest fluxes occurred during spring when discharge was high while the lowest values occurred in the fall and winter when discharge was low. However, seasonal changes in NEP did not reflect variations in river discharge. Discharge values decreased throughout the year while NEP rates fluctuated across seasons. On the other hand, NEP trends largely mirrored seasonal variations in air-water $CO_2$ fluxes. When the estuary acted as a source of $CO_2$, negative NEP was observed. In comparison, when the system served as a $CO_2$ sink, NEP was positive. From the annual mass balance model, the small difference between riverine input and export flux suggests that the majority of DIC*

*produced within the estuary is exchanged with the atmosphere rather than exported to the ocean. It is important to note that such conclusions were estimated based on surveys conducted during different months from different years. More research and data is needed to accurately ascertain seasonal variations in estuarine fluxes and NEP."*

*"Unlike in most previously studied estuaries, freshwater residence times in the Scheldt Estuary and Delaware Bay are generally long ranging from about one to a few months (Gay and O'Donnell, 2009; Borges and Abril, 2011). In contrast, the smaller stratified Randers Fjord has a much shorter residence time (few days) (Nielsen et al., 2001). In the smaller Randers Fjord, $CO_2$ emission to the atmosphere is lower than net community production (NCP) in the mixed layer or much less significant (Gazeau et al., 2005). This occurrence is partly due to the decoupling in ecosystem production caused by water stratification. As organic matter is produced in the surface waters, its degradation occurs in the bottom waters, and ultimately delaying $CO_2$ exchange with the atmosphere (Borges and Abril, 2011). Further, total DIC export to the Baltic Sea is higher than riverine DIC inputs to the Randers Fjord, suggesting that, due to the shorter freshwater residence times of systems, much of the DIC produced by net respiration is exported rather than removed to the atmosphere (Gazeau et al., 2005). Comparably, the Rhine exhibits extremely rapid freshwater residence time (~2 days) due to intense freshwater discharge (~2200 $m^3$ $s^{-1}$). Such rapid turnover time, leads to reduced emission of methane ($CH_4$) to the atmosphere by bacterial oxidation and smaller internal DIC production due to net heterotrophy (Borges and Abril, 2011). However, lateral inputs from intertidal marsh systems in small estuaries can enhance accumulation and degradation of organic matter in surface waters, resulting in high $CO_2$ degassing fluxes (Dai and Wiegert, 1996; Cai and Wang, 1998; Neubauer and Anderson, 2003). Due to the broad geographic size of the Delaware Bay, the effect from the production and decomposition of marsh plants on $CO_2$ flux dynamics in the system may not be as influential as in smaller estuaries except near the coastlines where tides regularly flush marsh boundaries (Joesoef et al., 2015). In the macrotidal Scheldt Estuary, long freshwater residence time typically leads to DIC accumulation in the water column (Abril et al., 2000; Borges et al., 2006). In addition, in both the Delaware and Scheldt estuaries, small differences between riverine input and export flux suggests that the majority of DIC produced within the estuary is exchanged with the atmosphere rather than exported to the ocean. While similar NEP values may be observed, the enrichment of DIC in estuarine waters and resulting $CO_2$ exchange with the atmosphere will be more intense in estuarine systems with long residence times versus estuaries with short residence times (Borges and Abril, 2011). Thus, we suspect that in estuaries with long freshwater residence times (i.e. the Delaware Estuary), much of the DIC produced by NEP is most likely removed to the atmosphere rather than exported to the sea."*

While the title is clear and appropriate and the paper is generally well-structured, there are quite a few places where the language is not as clear as it could be. In particular, the abstract could use a fairly substantial rewrite in that the authors' wording is often vague. For example, they say "Our data further suggest that DIC in the Schuylkill River can be substantially different from DIC in the Delaware River, and thus in any river system, tributary contributions must be considered when addressing DIC inputs to the estuary" – the second half of this sentence is so broad as to be essentially meaningless. I'm not sure they ever made a really compelling argument for why this might be important. I am sure there are several reasons why it could be important, but the authors should articulate their reasons for believing this to be important.

*Response: We agree that certain points in the abstract are rather vague and need further clarification. Specifically here if the tributary contribution is not recognized, the high TA and DIC observed in this section of the estuary would be mistakenly attributed to an internal source (rather is from a tributary with high TA and DIC). Thus, we have revised and modified several sections of the abstract. We expand on why it is important to consider tributary contributions when addressing input fluxes, clarify the significance of changes in DIC to TA ratio, and elucidate on the importance of river input and export fluxes to the DIC mass balance model. The following sections of the abstract now read as:*

*"The ratio of DIC to TA, an understudied but important property, is high (1.11) during high discharge and low (0.94) during low discharge, reflecting additional DIC input in the form of carbon dioxide ($CO_2$), most likely from organic matter decomposition rather than from other bicarbonate ($HCO_3^-$) inputs due to drainage basin weathering processes. Our data further suggest that TA and DIC in the Schuylkill River can be substantially different than TA and DIC values in the Delaware River. Thus, tributary contributions must be considered when attributing estuarine DIC sources to internal carbon cycle vs. external processes such as drainage basin mineralogy, weathering intensity, and discharge patterns. Long-term records of increasing alkalinity in the Delaware and Schuylkill river support global shifts toward higher alkalinity in estuarine waters over time. Annual DIC input flux to the estuary and export flux to the ocean are estimated to be $15.7 \pm 8.2 \times 10^9$ mol C $yr^{-1}$ and $16.5 \pm 10.6 \times 10^9$ mol C $yr^{-1}$, respectively, while net DIC production within the estuary including inputs from intertidal marshes is estimated to be $5.1 \times 10^9$ mol C $yr^{-1}$. The small difference between riverine input and export flux suggest that, in the case of the Delaware Estuary and perhaps other large coastal systems with long freshwater residence times, the majority of the DIC produced by biological processes is exchanged with the atmosphere rather than exported to the sea. Based on a DIC mass balance model, we concluded that annually the Delaware Estuary is a weak heterotrophic system ($-1.3 \pm 3.8$ mol C $m^{-2}$ $yr^{-1}$), which is in contrast to many highly heterotrophic smaller estuaries."*

One part of the interpretation of the data that was never really explained to my satisfaction was why DIC and TA wouldn't both be diluted under higher discharge and thus why the DIC:TA ratio would change with discharge. I suspect that there's a role of temperature in biotic production of CO2 in soils that has a different slope than the temperature dependence of weathering, or something along these lines. The authors' could do a more complete job of illuminating readers on the various factors contributing to the seasonal changes of DIC vs. TA to create a fuller picture and narrative about why they observe a changing DIC:TA ratio through the seasons. The importance of temperature in driving these changes is critical if this work is to have any bearing on predictive studies under future climate change.

*Response: We agree the temperature dependence is likely different in respiratory $CO_2$ production and in weathering production of DIC, but it will be hard to argue only from this point as such differences will also be enhanced or depressed during the wet and dry cycle. We believe our proposed simple mechanism --a hydrodynamic control—provides a more fundamental first order control. That is during the rainy season more stored $CO_2$ from organic matter respiration is flushed out of the drainage basin and less time is permitted for $CO_2$ degassing from creeks and rivers before entering the estuary. We first add an explanation in the Introduction about the*

*source of DIC (organic carbon respiration and weathering) and Alkalinity (weathering reactions only). Then we explain the hydrodynamic control more clearly in the Discussion.*

*We further agree that temperature and moisture may also play a role in seasonal changes in DIC vs. TA, particularly regarding the role of temperature and moisture on biotic production of $CO_2$ in soils. We have added a few sentences discussing the impact that temperature and moisture may have on DIC:TA ratios as shown below. With a shift towards increasing temperatures and frequency of episodic weathering events, it is critical that we continue to explore such issues to help understand the impact of future climate changes.*

*"If only influenced by the weathering of carbonate and silicate minerals, the ratio of DIC to TA remains close to unity (Cai et al., 2004). On the other hand, $CO_2$ production from soil organic matter respiration and imbalances between production and respiration along the aquatic continuum can increase DIC to TA ratios (Mayorga et al., 2005). Presumably, during the wet season and high discharge periods, more $CO_2$ from soil organic matter respiration stored in the drainage basin is brought along the river system while less $CO_2$ is lost to the atmosphere due to a faster transport. Additionally, extensive research has shown positive correlations between temperature and soil respiration (Singh and Gupta, 1977; Reich and Schlesinger, 1992). While more research is needed, we suggest that changes in the DIC to TA ratio at the freshwater end-member may reflect inputs of soil organic matter respiration due to seasonal variations in discharge, temperature, and moisture content. As the ratio of DIC to TA determines aquatic pH and the buffer capacity (Egleston et al., 2010), our observations indicate that variation of this ratio should be considered in future global carbon cycle models, in particular regarding how wet and drought cycles in future climate scenarios would affect coastal water acidification and how coastal waters will respond to a changing terrestrial carbon export (Reginer et al., 2013; Bauer et al., 2013)."*

The discussion of the lithology in the watersheds of each of the study rivers was a bit more detailed than needed, so some of this could be placed into supplemental material, or the text could just be shortened, with the same references. I don't think the detail adds anything to the understanding that one part of the watershed and its tributary contain more carbonate rocks than other parts. Again, why does this matter? (I'm not saying it doesn't, but tell us why you find it important.)

*Response: Agreed. This section is perhaps too detailed and could be significantly shortened. We have substantially revised section 4.2 'Influence of tributary mixing', shortening in certain areas, and expanding on the importance that drainage basin mineralogy has on the carbonate chemistry of regional watersheds. The section now reads as follows:*

*"River TA collected at the Schuylkill River was much higher than TA in the Delaware River near the Philadelphia region (Fig. 5). A compilation of historical data collected at two USGS stations in Philadelphia from 1940 to the present show that not only was alkalinity in the Schuylkill River negatively correlated with river discharge, but that during periods of low river discharge markedly high alkalinity was observed (Fig. 7A). Further, historical records agreed remarkably well with our alkalinity measurements. Over the past two decades, after low river discharge (<*

*100 m$^3$ s$^{-1}$) alkalinity reached from 1300 to 2500 µmol kg$^{-1}$, nearly two-fold greater than alkalinity values observed at the Trenton end-member (Fig. 7B).*

*The mineralogy of the Schuylkill River drainage basin may have a significant impact on TA patterns throughout the Delaware estuarine system. Geographically, the lower Schuylkill drainage basin extends through the Piedmont province, underlain by a mixture of limestone, shale, gneiss, schist, and dolomite, before discharging into the Coastal Plain province and the Delaware River (Stamer et al., 1985). Within this region, the Schuylkill River flows through the Valley Creek basin in which 68% of the region is comprised of carbonate rocks (Sloto, 1990). The center of the basin, otherwise known as Chester Valley, is primarily underlain by easily eroded limestone and dolomite bedrock with regional flow discharging into the Schuylkill River. Thus, it is likely that high riverine TA in the Schuylkill River is due to the weathering of carbonate rocks in the lower Schuylkill drainage basin. We suggest that elevated DIC and TA values exhibited in the Delaware River near Philadelphia are the result of the mixing of relatively high carbonate freshwater from the Schuylkill River, specifically due to the chemical weathering of limestone and dolomite bedrock across the lower Piedmont province. In turn, tributary contributions must be considered when addressing total riverine DIC and TA fluxes as differences in drainage basin mineralogy can have a substantial impact on the carbonate chemistry throughout regional watersheds. Influences from human activities such as wastewater discharge, agriculture, and acid mine drainage may also contribute to the high TA, an issue that deserves further study (Raymond and Cole, 2003; Raymond et al., 2008)."*

**Minor Comments**

The language is fluent and mostly clear, save in a few places where the language becomes vague/imprecise. It may seem redundant to the writer, but there are numerous places where a few more words added would make the difference between vagueness and clarity.

P1, L7: "widely understudied" seems like a bit of a non sequitur/oxymoron.

*Changed to 'understudied'.*

P1, L26: You might want to say "flux to the coastal ocean" instead of just "flux to the ocean" as in coastal carbon cycle circles, we also discuss export from coastal oceans to the open ocean.

*Good point. We have changed to 'flux to the coastal ocean'.*

P2, L2: "majority of the DIC produced" – in the estuary, I presume? Clarify.

*Changed to 'majority of the DIC produced in the estuary'.*

P2, L10: land-to-ocean continuum?

*Correct.*

P2, L 25: "The supply of inorganic carbon by rivers: : :" – to the coastal ocean?

*Yes, we have changed to 'The supply of inorganic carbon by rivers to the coastal ocean'.*

P2, L 23-24: I thought it was just for carbonate minerals that the $CO_2$ is eventually released back to the atmosphere via oceanic carbonate sedimentation, stoichiometrically speaking. Please verify that this statement is correct.

*Yes, the reviewer is correct. During $CaCO_3$ weathering, $CO_2$ is removed from the atmosphere while at sea this process is reversed during $CaCO_3$ precipitation. However, for silicate weathering $CO_2$ is removed from the atmosphere but this process cannot be reversed as diatoms precipitate opal minerals (no C in it). Only after a much slower process later (reverse weathering that converts $CaCO_3$ and opal minerals to silicate minerals) is the cycle completed. However, as that is beyond our research here, we do not mention it but only cite the Lerman paper. We have modified the sentence as follows:*

*"The weathering of carbonate and silicate minerals consumes atmospheric $CO_2$ and transports $HCO_3^-$ ions and subsequent cation and anion products into oceanic systems. Eventually, $CO_2$ is released back into the atmosphere via oceanic carbonate sedimentation and volcanic activity (Lerman et al., 2004; Regnier et al., 2013)."*

In several places, the authors use the word "impact" when "affect" would be more appropriate. "Impact" is often used to convey negative connotations.

*Good point. We have changed accordingly.*

P3, L3: "by weathering and decomposition"

*Changed.*

P3, L12: "more large bay systems" would be clearer

*Agreed. We have changed to 'an urgent need to expand global research to more large bay systems'.*

P4, L 18: replace "ongoing" with "underway"

*Replaced with 'underway'.*

P4, L27-28: does this method of filtering samples affect the DIC values? I presume the references given address this, but if not, it would be good for these authors to address whether filtering samples introduces any artifacts or bias. Filtering DIC samples is not typical (e.g., per the Dickson et al. 2007 SOPs for the CO2 system), but can be done without introducing bias with adequate care (e.g. Bockmon and Dickson 2015? L&O).

*Correct. We did not filter samples unless they were collected in the upper tidal river portion of the estuary which was heavily turbid. We have revised the sampling description.*

P5, L4: Are you sure it's precision that is +/- 2 umol/kg? Vs. some overall uncertainty or average offset from CRMs?

*Good point. It is the overall uncertainty of our measurements with respect to the CRMs. We have changed to, "All measurements were calibrated against certified reference material (provided by A.G. Dickson from Scripps Institution of Oceanography) with an uncertainty of ± 2 μmol kg$^{-1}$ (Huang et al., 2012)."*

P5, L 5-7: Need to state pH scale is NBS.

*Agreed. Added pH scale in NBS.*

P6, L 10: "northernmost"

*Changed.*

P6, L21-22: might be good to clarify that this is from the rivers where measurements were taken (vs. the scaled up estimate presented later on).

*Good point. We now clarify and add the Delaware, Schuylkill, and Christina rivers to this description.*

P6, L27-28: re-cite figure here? Here and just below, it seems like there are a few steps left out of your description of how you did the calculations.

*We have re-cited the figure and have expanded on the description of our calculations as follows:*

*"The effective river end-member concentrations of DIC and TA were calculated by extrapolating the DIC and TA conservative mixing lines from the high salinity waters to zero salinity (Fig. 3) (Cai et al., 2004; Guo et al., 2008). The difference between the effective and actual concentrations at the river end-member indicates the amount of DIC and TA added or removed during mixing and therefore not transported to the ocean (Boyle et al., 1974; Cai and Wang, 1998; Liu et al., 2014). Using the effective concentrations and the combined river discharge for the Delaware, Schuylkill, and Christina rivers recorded in each cruise period plus the average discharges measured during the month prior to each survey, we estimate net DIC and TA export fluxes for each cruise (Table 3)."*

P7, L5: Not enough info given about what this data set is and how the data compare to yours. Put in methods or otherwise describe.

*Agreed. We now elaborate on the specific USGS alkalinity parameter codes used and compiled this information into a new table (Table 4). We also expand on the importance of historical USGS water quality data as shown below.*

*"The extensive and routine collection of water samples conducted by USGS allows us to explore long term trends (from the mid-20$^{th}$ to early 21$^{st}$ century) in alkalinity and discharge in the Delaware and Schuylkill rivers (USGS stations 01463500 and 01474500, respectively). For USGS alkalinity values, we use similar approaches as conducted in Stets et al., 2014. We combine 8 various parameter codes that include alkalinity, acid neutralizing capacity (ANC), or $HCO_3^-$ (Table 4). Alkalinity and ANC follow identical electrometric procedures except that alkalinity samples are filtered while ANC samples are not."*

Table 4. USGS parameter codes used during analysis

| Parameter Code | Parameter Description | Total Count | Percentage of Total Count |
|---|---|---|---|
| 00410 | Acid neutralizing capacity, water, unfiltered, fixed endpoint titration, field | 920 | 28.5 |
| 00419 | Acid neutralizing capacity, water, unfiltered, inflection-point titration, field | 25 | 0.8 |
| 00440 | Bicarbonate, water, unfiltered, fixed endpoint titration, field | 1529 | 47.4 |
| 00450 | Bicarbonate, water, unfiltered, inflection-point titration, field | 25 | 0.8 |
| 00453 | Bicarbonate, water, filtered, inflection-point titration, field | 86 | 2.7 |
| 29801 | Alkalinity, water, filtered, fixed endpoint titration, laboratory | 133 | 4.1 |
| 39086 | Alkalinity, water, filtered, inflection-point titration, field | 283 | 8.8 |
| 90410 | Acid neutralizing capacity, water, unfiltered, fixed endpoint titration, laboratory | 224 | 6.9 |

P7, L7: This seasonality doesn't agree with what you described above (summer+fall vs. spring+summer, etc.).

*Good catch. We have corrected it to 'TA was highest during low flow season (fall) and lowest during high flow season (spring)'.*

P7, L16: "strong" correlations, not "high" (or "highly correlated")

*Changed to 'strong'.*

P7, L20: "dilution of weathering products" (vs. production)

*Agreed. We have changed to 'products'.*

P8, L3: could be faster transport or lower surface area to volume ratio (i.e. deeper)

*Good point. We have added this detail.*

*"Presumably, during the wet season and high discharge periods, more $CO_2$ from soil organic matter respiration stored in the drainage basin is brought along the river system while less $CO_2$ is lost to the atmosphere due to a faster transport and lower surface area to volume ratio (i.e. deeper water depths)."*

P9, L1: I am not sure what you mean by "physiographic"

*We have significantly modified and shortened this section as requested in your earlier comments and by others. This sentence has been removed and is no longer in the discussion.*

P9, L7: not totally clear what "the historical record" refers to - all USGS data? Just a subset?

*Yes, we agree that before this was unclear throughout the manuscript. We have added more detail to the exact USGS data used for our analysis and it is now described at the beginning of section 4.1 'Influence of river discharge and weathering intensity' and in the addition of Table 4 as described in the above responses.*

P9, L8: closer to 3 decades, at 26 years – maybe "over recent decades" is better?

*Agreed. We have changed to 'over recent decades'.*

P9, L9-11: to facilitate reading the paper, it may be best to stick to river names rather than mixing in city names for those readers outside your region.

*Yes, we now refer to the river names instead of city names.*

P9, L 16-18: After too much detail on watershed lithology at the start of this section, a bunch of things are summarily mentioned without discussing how these processes might contribute to TA change sufficiently (e.g. would these processes individually increase or decrease TA, and how?).

*We have significantly modified and shortened this section as requested in your earlier comments and by others. Please refer to the revised 4.2 section as described in our previous responses.*

P9, 26-31: See previous – superficial treatment of these factors (also "can also have huge effects" on following page). Be more specific about the relative roles each of these factors would play if they are important.

*Yes, we have expanded greatly on section 4.3 'Historical trends in riverine alkalinity'. Further, we now refer to several more comprehensive studies that have explored long term alkalinity records across various streams and watersheds throughout the U.S. (Kaushal et al., 2013; Stets et al., 2014). We also expand on how various factors effect long term alkalinity patterns as described below:*

*"A more comprehensive study conducted by Kaushal et al., 2013 examined long-term trends in river alkalinity at 97 different stream and river locations throughout the eastern U.S. They observed increasing alkalinity trends at 62 of the 97 river locations (64%). Moreover, of the remaining sites, none showed any statistically decreasing alkalinity trends. Various contributing factors can influence long-term trends in river alkalinity such as carbonate lithology, acid deposition, and topography in watersheds. Kaushal et al., 2013 suggests that increased acid deposition elevates riverine alkalinity by promoting weathering processes, particularly in watersheds with high carbonate lithology. Further, watershed elevation may be a good predictor for alkalization rates. Acid deposition may be greater at higher elevations, and such areas tend to have thinner soils and a weaker buffering capacity, increasing susceptibility to the effects of acid deposition. Recent studies show that human induced land-use changes such as deforestation, agricultural practices (Oh and Raymond, 2006), and mining activities (Brake et al., 2001; Raymond and Oh, 2009) have direct impacts on the buffering capacity of streams and rivers. Through chemical weathering processes, enhanced precipitation and local runoff can also have huge effects on increased alkalinity in coastal ecosystems (Raymond et al., 2008). For example, over the past century, total alkalinity export from the Mississippi River to the Gulf of Mexico has risen by nearly 50% due to widespread cropland expansion and increased precipitation in the watershed (Raymond and Cole, 2003; Raymond et al., 2008). Comparably, Stets et al., 2014 explored historical time series of alkalinity values in 23 different riverine systems throughout the U.S. They found increasing alkalinity trends at 14 of these locations with the majority occurring in the Northeastern, Midwestern, and Great Plains areas of the U.S. While most sites observed increasing alkalinity values with time, decreasing trends were found in the Santa Ana, Upper Colorado, and Brazos rivers. Factors contributing to decreasing trends at these locations include dilution by water from external sources outside the basin and retention of weathering products in storage reservoirs."*

P10, L18-20: this is very qualitative. Can you be more quantitative about this?

*We have added the following quantitative details:*

*"In March and August 2014, $pCO_2$ was low (160 – 350 μatm) and $CO_2$ take was greatest (-21 – 2.5 mmol $m^{-2}$ $d^{-1}$) throughout the mid- and lower bay regions, indicating biological $CO_2$ removal (Joesoef et al., 2015)."*

P11, L12-17: This sentence seems circular to me – how are you defining the difference between input and inflow? If you mean to consider groundwater inputs too, you need to be more concrete and specific with your wording. (Also, there was the roughly 10% from wastewater treatment plants [WWTPs] from up top not mentioned here. Intentional?)

*Yes, we meant to define as 70% of the freshwater 'input', not 'inflow' and have changed this. Although minor, we have added inputs from WWTPs to the remaining 30% percentage (from small rivers and nonpoint source runoff). We also acknowledge that bicarbonate concentrations from these remaining sources may not be the same as the three main Delaware river systems. However, since additional research and data is needed to accurately determine their contribution, we assume that the remaining 30% yield similar concentrations. We now acknowledge this uncertainty as follows:*

*It is important to note that these final flux values are strictly a rough estimate. We acknowledge that average riverine DIC and TA concentrations from remaining small rivers and nonpoint source runoff are not necessarily equivalent to the weighted DIC and TA averages for the Delaware, Schuylkill, and Christina rivers. As such uncertainties are most often neglected, it is imperative to consider their effect on final flux estimates. However, since additional research and data collection is needed, here we assume that the mineralogy and drainage basins of the remaining 30% yield similar carbonate concentrations as Delaware's three major river systems.*

P11, L26: Do you mean water column/internal estuarine CO2 production (per top of next page, I think you do)? Calling it "production" without further clarification of what is being produced gets confusing when primary/community production may also be involved.

*Agreed. We have changed this to 'internal estuarine $CO_2$ production'.*

P12, L5: near the top, you had a figure for 30 m^3/s from WWTPs – this seemed like not a trivial part of the total input.

*We agree that inputs from WWTPs is important and can influence river carbonate concentrations and overall metabolic processes, especially in the upper tidal river. However, much more research is needed near waste water discharge locations and treatment plants to evaluate their impact on the Delaware river system. In turn, for simplicity we ignore WWTP contributions in our DIC mass balance model.*

P12, L27: "intertidal" instead of "internal," yes?

*Correct. We have changed to 'intertidal'.*

P13, L9: "Here" – where are you referring to?

*We have changed to 'In these small river systems with rapid residence times' to clarify what we are referring to.*

P14, L1: "compared to total DIC input flux" – suggest adding "from rivers"

*Agreed. The phrase now reads as 'compared to total DIC input flux from rivers'.*

P26: Seems to ignore interannual variability to list cruises by month w/o noting they occurred in different years. Do you have enough data on interannual variability to justify that this makes more sense than an alternative? (I don't feel strongly that this shouldn't be done but am curious about the choice to do it this way – would be nice to have some explanation – but doesn't need to be extensive).

*This is an interesting thought and we have now pointed this issue out in section 4.6 to inform the audience. We have added the following sentences to clarify this issue:*

*"Discharge values decreased throughout the year while NEP rates fluctuated across seasons. On the other hand, NEP trends largely mirrored seasonal variations in air-water $CO_2$ fluxes. When the estuary acted as a source of $CO_2$, negative NEP was observed. In comparison, when the system served as a $CO_2$ sink, NEP was positive. From the annual mass balance model, the small difference between riverine input and export flux suggests that the majority of DIC produced within the estuary is exchanged with the atmosphere rather than exported to the ocean. It is important to note that such conclusions were estimated based on surveys conducted during different months from different years. More research and data is needed to accurately ascertain seasonal variations in estuarine fluxes and NEP."*